# PLAN AND BUDGET: EFFECTIVE AND EFFICIENT TEST-TIME SCALING ON LARGE LANGUAGE MODEL REASONING

**Junhong Lin**[1]* **Xinyue Zeng**[2]* **Jie Zhu**[2] **Song Wang**[3]
**Julian Shun**[1] **Jun Wu**[4] **Dawei Zhou**[2]

[1]MIT CSAIL   [2]Virginia Tech   [3]University of Virginia   [4]Michigan State University

{junhong,jshun}@mit.edu   {xyzeng,jiez19,zhoud}@vt.edu
sw3wv@virginia.edu   wujun4@msu.edu

## ABSTRACT

Large Language Models (LLMs) have achieved remarkable success in complex reasoning tasks, but their inference remains computationally inefficient. We observe a common failure mode in many prevalent LLMs, *overthinking*, where models generate verbose and tangential reasoning traces even for simple queries. Recent work has tried to mitigate this by enforcing fixed token budgets, however, this can lead to *underthinking*, especially on harder problems. Through empirical analysis, we identify that this inefficiency often stems from unclear problem-solving strategies. To formalize this, we develop a theoretical model, **BAM** (**B**udget **A**llocation **M**odel), which models reasoning as a sequence of sub-questions with varying uncertainty, and introduce the $\mathcal{E}^3$ metric to capture the trade-off between correctness and computation efficiency. Building on theoretical results from BAM, we propose **PLAN-AND-BUDGET**, a model-agnostic, test-time framework that decomposes complex queries into sub-questions and allocates token budgets based on estimated complexity using adaptive scheduling. PLAN-AND-BUDGET improves reasoning efficiency across a range of tasks and models, achieving up to **70%** accuracy gains, **39%** token reduction, and **193.8%** improvement in $\mathcal{E}^3$. Notably, it improves the efficiency of a smaller model (DS-Qwen-32B) to match the efficiency of a larger model (DS-LLaMA-70B), demonstrating PLAN-AND-BUDGET's ability to close performance gaps without retraining. Our code is available at Plan-and-Budget.

## 1 INTRODUCTION

Large Language Models (LLMs) exhibit strong generalization capabilities, enabling them to perform a wide range of tasks, such as mathematical problem solving (Ahn et al., 2024; Imani et al., 2023), scientific question answering (Huang et al., 2024; Lu et al., 2022a), and structured reasoning (Guo et al., 2025; Wei et al., 2022), without task-specific retraining. Recent advances in test-time computation, such as Chain-of-Thought (CoT) prompting (Wei et al., 2022), self-consistency (Wang et al., 2023), and tool-augmented inference (Chen et al., 2023), have significantly improved their performance on complex, multi-step reasoning tasks. These enhancements have paved the way for LLMs to become increasingly deployed in high-stakes domains such as education (Golshan & Academy, 2023), finance (Wang et al.), law (Katz et al., 2024), and scientific research (Taylor et al., 2022), where robust reasoning at inference time is critical.

Despite these advances, deploying LLMs in real-world settings introduces new challenges, particularly in scenarios that require deliberative reasoning under computational and time constraints. A key issue is the lack of calibrated reasoning behavior during inference. Although LLMs are proficient in multi-step reasoning, they often struggle to regulate how much reasoning effort is appropriate for a task. This miscalibration manifests in two major failure modes: *overthinking* (Sui et al.; Chen et al., 2025; Turpin et al., 2023), where models generate unnecessarily long and tangential reasoning paths,

---

*Equal contribution.

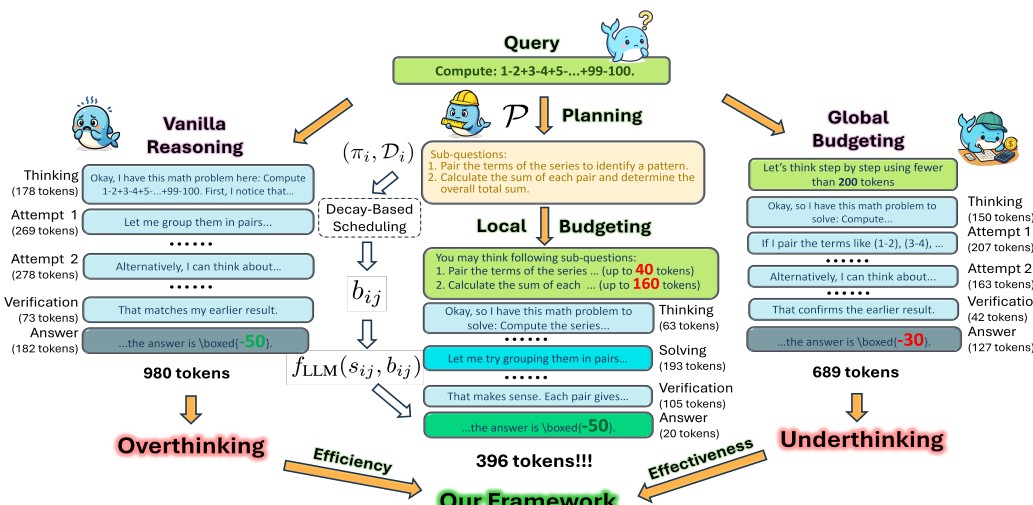

Figure 1: Illustration of REASONING MISCALIBRATION. Vanilla reasoning results in overthinking and wastes tokens; global budgeting results in underthinking and fails. Our method combines planning and local budgeting to guide structured, efficient reasoning, achieving the correct answer with fewer tokens.

even for simple queries, incurring excessive computational cost without improving accuracy; and *underthinking* (Wang et al., 2025a; Wei et al., 2022), where models terminate reasoning prematurely, sacrificing correctness to conserve resources. Recent methods (Lee et al., 2025; Xu et al., 2025; Han et al., 2025) have attempted to mitigate overthinking by introducing hard token constraints (e.g., "using fewer than $B$ tokens" in the prompt). While these strategies may be effective on simpler tasks, they often degrade performance on complex queries by inducing underthinking, highlighting the limitations of fixed, non-adaptive approaches. To the best of our knowledge, there is limited work that has systematically addressed both overthinking and underthinking in a unified framework.

In this paper, we take the first step toward closing this gap. With a comprehensive empirical study of test-time reasoning behavior in state-of-the-art LLMs ranging from 32B to 70B parameters, we uncover a pervasive phenomenon that we term **REASONING MISCALIBRATION**, a failure mode where models exhibit unregulated inference depth during reasoning. This miscalibration manifests as either overthinking, where the model engages in unnecessary and tangential reasoning, or underthinking, where reasoning terminates prematurely. Our study reveals that reasoning miscalibration is frequently triggered by two types of queries: (1) trivial-but-ambiguous queries, which elicit diffuse token distributions and lead to speculative reasoning; and (2) hard-and-rare queries, where models engage in shallow trial-and-error without meaningful convergence. These findings raise a central research question: ***How can we characterize the internal reasoning and inference mechanisms of LLMs, and how can we guide them to allocate computation adaptively based on task complexity?***

To answer this question, we reinterpret reasoning miscalibration as a resource allocation problem under uncertainty. For trivial-but-ambiguous queries, although the underlying solution is simple, the model may follow multiple semantically equivalent reasoning paths, leading to redundant exploration and excessive inference depth. In contrast, for hard-and-rare queries, genuine uncertainty arises from fragile intermediate conclusions or incomplete internal knowledge, yet under fixed budgets, the model may prematurely commit to incomplete reasoning trajectories. These contrasting behaviors suggest that reasoning failures do not arise merely from too little or too much computation, but from a mismatch between computational effort and the uncertainty structure of the problem. Some intermediate sub-questions require substantial deliberation because ambiguity at these stages can alter subsequent reasoning trajectories and ultimately affect correctness, while others no longer have ambiguity at these stages and further reasoning yields negligible benefit. Motivated by this perspective, we introduce the **Budget Allocation Model (BAM)**, a theoretical framework that characterizes reasoning as a sequence of sub-questions with heterogeneous uncertainty reduction dynamics. BAM formalizes how a fixed computational budget should be distributed across sub-questions so as to prioritize steps where uncertainty can be most effectively reduced. From this viewpoint, effective

reasoning follows two principles: (1) reasoning should be *structured*, decomposing complex queries into targeted sub-questions; and (2) computation should be *adaptive*, allocating more budget to sub-questions with higher uncertainty.

Building on BAM, we propose a novel compute-efficient reasoning strategy, called **PLAN-AND-BUDGET**, which consists of two stages: **Plan** and **Budget**. In the **Plan** Step, the model decomposes the original query into a sequence of sub-questions, providing a soft scaffold for structured reasoning. In the **Budget** Step, we apply simplified scheduling strategies that dynamically assign token budgets to each sub-question, guided by its uncertainty pattern, following the BAM principle. To evaluate our approach, we introduce $\mathcal{E}^3$, the **E**fficiency-Aware **E**ffectivenss **E**valuation Score, which captures the trade-off between reasoning accuracy and computational cost. Unlike conventional efficiency metrics that overlook output quality, $\mathcal{E}^3$ offers a more robust measure of inference performance.

We evaluate our method through extensive experiments across four state-of-the-art LLMs, including DeepSeek-R1 Distill-Qwen-32B (DS-Qwen-32B) (Guo et al., 2025), QwQ-32B (Team, 2025), DeepSeek-R1 Distill-Llama-70B (DS-LLaMA-70B) (Guo et al., 2025), and OpenAI o4-mini (OpenAI, 2025) on three representative task domains: mathematical reasoning, instruction following, and agentic planning. Our method is model-agnostic—it requires no retraining or fine-tuning, relying only on prompting and lightweight planning. Despite this simplicity, PLAN-AND-BUDGET consistently improves all four LLMs across all benchmarks. We observe downstream accuracy gains of up to **70%**, token usage reductions of up to **39%**, and combined efficiency improvements (as measured by $\mathcal{E}^3$) of up to **193.8%** over strong baselines. An especially notable case comes from the agentic planning task domain, where a smaller DS-Qwen-32B improves from a low $\mathcal{E}^3$ of 0.16 to 0.47 using PLAN-AND-BUDGET, **closing the gap with the larger DS-LLaMA-70B model** ($\mathcal{E}^3 = 0.50$) without planning. This demonstrates that uncertainty-guided planning and budgeting can act as inference-time equalizers, boosting the efficiency and competitiveness of smaller models without retraining. Together, these findings underscore the promise of principled compute allocation for more calibrated, efficient, and accessible LLM inference.

## 2 RELATED WORK AND PRELIMINARIES

### 2.1 RELATED WORK

**Scaling Laws.** Recent work has explored how test-time computation affects LLM performance, showing that an increased inference budget can reduce failure rates but often suffers from diminishing returns (Snell et al., 2024; Wu et al., 2025; Zeng et al., 2025; Dai et al., 2025). Methods like MCTS-Judge (Wang et al., 2025b) and EAG (Mei et al., 2025) demonstrate the benefits of adaptive computing in tasks like code evaluation and multi-hop reasoning. Unlike prior work focusing on simply increasing compute, we investigate how to allocate it efficiently through structured planning and uncertainty-aware budgeting.

**Uncertainty.** Quantifying uncertainty in deep models is often framed through epistemic vs. aleatoric components (Hüllermeier & Waegeman, 2021), with techniques like MC Dropout (Gal & Ghahramani, 2016), ensembles (Lakshminarayanan et al., 2017), and evidential learning (Huang et al., 2021). Recent work extends these ideas to LLMs via consistency checks and parameter-efficient ensembles (Lu et al., 2022b; Tonolini et al., 2024; Mühlematter et al., 2024; Ouyang et al., 2024). Our work builds on this by using uncertainty decomposition to guide token allocation at inference time, offering a novel application of uncertainty for test-time efficiency.

### 2.2 PRELIMINARIES

We begin by summarizing previous work and introducing the key notations used throughout this work. Table 1 lists the symbols relevant to our reasoning formulation.

Table 1: Notation Summary

| Symbol | Description |
|---|---|
| $m$ | Number of sub-questions |
| $x_i$ | $i$-th query |
| $s_{ij}$ | $j$-th sub-question of query $x_i$ |
| $b_{ij}$ | Tokens allocated to sub-question $s_{ij}$ |
| $\beta_{ij}$ | Complexity of sub-questions $s_{ij}$ |
| $B$ | Total token budget per query |
| $\pi_i$ | Decomposition plan for query $x_i$ |
| $w_{ij}$ | Normalized complexity weight for $s_{ij}$ |
| $\gamma, \epsilon, p$ | Decay scheduler hyperparameters |
| $c_{ij}$ | Parameter characterizing epistemic uncertainty reduction |

**Reasoning Miscalibration in LLMs.** While LLMs excel at complex reasoning tasks, they often struggle to regulate how much inference effort is appropriate per query. We refer to this phenomenon as REASONING MISCALIBRATION. It describes a mismatch between task complexity and the depth of reasoning a model performs at test time.

This miscalibration presents itself in two primary modes: (1) Overthinking (Sui et al.; Chen et al., 2025; Turpin et al., 2023), where the model engages in excessively verbose or tangential reasoning even for simple queries, incurring unnecessary computational cost and introducing noise or contradictions; and (2) Underthinking (Wang et al., 2025a; Wei et al., 2022), where the model prematurely stops reasoning to conserve budget, often yielding incomplete or incorrect answers.

Contrary to the common belief that allocating more decoding tokens leads to better performance, we observe that excessive generation can degrade quality. In our empirical analysis, we show that longer outputs can lead models to wander within the solution space, becoming verbose, redundant, or self-inconsistent. Our findings suggest that REASONING MISCALIBRATION does not stem from a lack of knowledge or model capacity, but rather from the model's inability to dynamically align reasoning effort with a query's evolving informational needs—particularly in response to uncertainty at each step. We leverage a foundational concept of predictive uncertainty (Hüllermeier & Waegeman, 2021) which decomposes the total uncertainty $\mathcal{U}(x)$ for a given input $x$ into two distinct components:

$$\mathcal{U}(x) = \mathcal{U}_{\text{epistemic}}(x) + \mathcal{U}_{\text{aleatoric}}(x),$$

where $\mathcal{U}_{\text{epistemic}}(x)$ captures uncertainty due to incomplete knowledge (and is reducible through targeted computation), while $\mathcal{U}_{\text{aleatoric}}(x)$ accounts for irreducible ambiguity or noise in the input. Recent work by Falck et al. (2024) extends this decomposition to LLMs, revealing that LLMs display dynamic uncertainty profiles throughout inference. These evolving patterns offer valuable insights into both the models' reasoning processes and the quality of their generated outputs. We further demonstrate the validity of this decomposition in the LLM setting through formal analysis in Appendix C.

**Problem Definition.** In multi-step reasoning, the decomposition above reveals a crucial insight: REASONING MISCALIBRATION arises from unregulated computational effort across sub-questions with varying uncertainty levels. Some sub-problems demand greater inference depth to reduce epistemic uncertainty, while others, dominated by aleatoric uncertainty, benefit from early termination or concise solutions. Yet current LLMs lack a mechanism to adaptively allocate computation across these stages. This misalignment leads to inefficiency and degraded reasoning quality. Our goal is to improve efficiency while mitigating REASONING MISCALIBRATION.

**Efficiency-Aware Effectiveness Evaluation: $\mathcal{E}^3$ Score.** We introduce the $\mathcal{E}^3$ index as an efficiency-aware metric that jointly captures reasoning quality and computational cost. Rather than treating token usage and accuracy as separate concerns, $\mathcal{E}^3$ directly quantifies their trade-off:

$$\mathcal{E}^3 = A \cdot \frac{A}{T} = \frac{A^2}{T}.$$

Here, $A$ denotes the average accuracy achieved across a set of queries, and $T$ represents the average number of decoding tokens used per query. Earlier work typically measures efficiency as accuracy per token (Muennighoff et al.; Lee et al., 2025) (i.e., $A/T$). By putting more weight on accuracy, $\mathcal{E}^3$ emphasizes correctness, discouraging degenerate strategies that minimize token usage at the cost of quality. In doing so, it reflects how well a model aligns its computational effort with task complexity, rewarding those who invest more where needed and conserving resources otherwise. Thus, the $\mathcal{E}^3$ score provides a principled evaluation framework for assessing whether a model mitigates REASONING MISCALIBRATION while maximizing reasoning efficiency. To formally characterize this efficiency-accuracy trade-off under adaptive computation, we now define our target optimization problem as follows.

**Problem 1.** *LLM Reasoning Calibration*
*Given: (1) A set of complex queries $\{x_1, \ldots, x_n\}$, where each $x_i$ can be decomposed into a sequence of $m$ sub-questions; and (2) A total token budget $B_i$ for each query $x_i$.*
*Find: A computation strategy that maximizes the efficiency-aware score $\mathcal{E}^3 = \frac{A^2}{T}$, subject to the constraint $B_i$ for each query. The objective is to allocate inference effort in a way that prioritizes correctness under limited computational resources.*

## 3 BUDGET ALLOCATION MODEL (BAM)

To address reasoning miscalibration in Problem 1, we need a principled method for allocating computation across sub-questions with varying uncertainty. As established by Falck et al. (2024), effective reasoning requires focusing effort where epistemic uncertainty is high, and limiting it where aleatoric noise dominates. Existing methods lack a formal mechanism for this adaptive allocation. They often treat all reasoning steps uniformly, leading to inefficient budget use and exacerbating reasoning miscalibration. To bridge this gap, we introduce the **Budget Allocation Model (BAM)**, a theoretical framework that models token allocation as uncertainty reduction under a fixed budget. BAM provides a principled foundation for our adaptive reasoning framework presented in Section 4.

To distribute a finite token budget $B_i$ across the sub-questions of $x_i$, we adopt a Bayesian decision-theoretic formulation that aims to maximize reasoning utility by minimizing total uncertainty. While standard LLM inference is deterministic, recent theoretical work suggests that in-context learning can be viewed as implicit Bayesian inference (Falck et al., 2024). We leverage this view to characterize reasoning behavior, even without performing explicit posterior sampling at test time. We assume an inverse power law governs epistemic uncertainty reduction for sub-question $s_{ij}$ with token allocation $b_{ij}$:

$$\mathcal{U}_{\text{epistemic}}(s_{ij} \mid b_{ij}) = \frac{c_{ij}}{b_{ij}^{\beta_{ij}}}, \tag{1}$$

where $c_{ij} > 0$ reflects the initial epistemic uncertainty and $\beta_{ij} \geq 1$ captures the complexity of reducing that uncertainty (higher $\beta_{ij}$ corresponds to being easier to reduce the uncertainty). This formulation is motivated by established Neural Scaling Laws (Kaplan et al., 2020; Hoffmann et al., 2022; Zeng et al., 2025), which demonstrate that model loss—a proxy for uncertainty—scales as a power law with compute. We extend this perspective to test-time reasoning by modeling the reduction of inference error as a power-law function of token allocation, where additional computation yields diminishing returns (Snell et al., 2024; Wu et al., 2025).

We model total uncertainty as the sum of the epistemic and aleatoric components:

$$\mathcal{U}(s_{ij} \mid b_{ij}) = \frac{c_{ij}}{b_{ij}^{\beta_{ij}}} + \mathcal{U}_{\text{aleatoric}}(s_{ij}). \tag{2}$$

Here, we treat $\mathcal{U}_{\text{aleatoric}}$ as a constant with respect to $b_{ij}$, since it reflects irreducible uncertainty that cannot be mitigated through additional inference effort. A proof of the decomposition of total uncertainty in LLMs is provided in Section C.

We define the utility of resolving a sub-question $s_{ij}$ as a decreasing function of its uncertainty:

$$r(s_{ij} \mid b_{ij}) = \alpha \cdot (1 - \mathcal{U}(s_{ij} \mid b_{ij})), \tag{3}$$

where $\alpha$ is a model/task-based scaling factor. The total utility for query $x_i$ is then:

$$\mathcal{R}_{\text{total}} = \sum_{j=1}^{m} r(s_{ij} \mid b_{ij}). \tag{4}$$

The optimal budget allocation solves the following constrained optimization problem:

$$\max_{b_{i1},\ldots,b_{im}} \sum_{j=1}^{m} \alpha \cdot \left(1 - \frac{c_{ij}}{b_{ij}^{\beta_{ij}}} - \mathcal{U}_{\text{aleatoric}}(s_{ij})\right) \quad \text{s.t.} \quad \sum_{j=1}^{m} b_{ij} \leq B_i. \tag{5}$$

By introducing a Lagrange multiplier $\lambda$ to handle the budget constraint and solving the resulting Lagrangian, we arrive at the optional solution:

$$b_{ij} = B_i \cdot \frac{(c_{ij}\beta_{ij})^{\frac{1}{\beta_{ij}+1}}}{\sum_k (c_{ik}\beta_{ik})^{\frac{1}{\beta_{ik}+1}}}. \tag{6}$$

This allocation rule reveals a unimodal relationship between $b_{ij}$ and $\beta_{ij}$, i.e., token budget increases with complexity up to the peak, then decreases as further effort yields diminishing returns. This relationship is key to mitigating reasoning miscalibration: *moderately difficult sub-questions receive more tokens to avoid underthinking, while overly difficult ones receive fewer to prevent overthinking*. BAM thus provides a principled, self-regulating mechanism for aligning inference effort with reasoning value. Detailed proofs are provided in Appendix D and E.

## 4 REASONING CALIBRATION FRAMEWORK: PLAN-AND-BUDGET

Building directly on BAM's principle in Eq. 6, the optimal allocation distributes a query-level budget $B$ across sub-questions by maximizing expected uncertainty reduction. In practice, however, the marginal uncertainty reduction curves are unknown, and the parameters $c_{ij}$ and $\beta_{ij}$ in Section 3 are not directly observable in black-box LLMs. Directly estimating these quantities would require expensive sampling, defeating our efficiency goal.

Instead, we propose **PLAN-AND-BUDGET** as a practical instantiation that operationalizes the structural insight of BAM using a lightweight surrogate. Recall that BAM prescribes allocating more computation to sub-questions with greater potential for uncertainty reduction. While the exact marginal gains are inaccessible, we observe that in multi-step reasoning tasks, early stages (e.g., problem interpretation and strategy formation) typically exhibit higher uncertainty than later refinement steps. Motivated by this qualitative structure, we adopt decay-based scheduling strategies that front-load computational budget toward earlier reasoning levels. Polynomial and cosine decay, in particular, approximate the non-uniform allocation shape predicted by BAM, allocating larger budgets to high-uncertainty stages while gradually reducing allocation as reasoning stabilizes.

This schedule is budget-feasible by construction, ensuring that the total allocation respects the global constraint while preserving the non-uniform allocation structure implied by BAM.

### 4.1 PLANNING STEP: QUESTION DECOMPOSITION AS GUIDED SCAFFOLD

Inspired by human problem-solving strategies, we use query decomposition as a reasoning scaffold to improve efficiency and focus. Our planning process has two phases:

**Phase 1: Automatic Planning.** A lightweight planning function $\mathcal{P}$ decomposes $x_i$ into an ordered sequence of sub-questions $\pi_i$ and their estimated complexity scores $\mathcal{D}_i$:

$$\mathcal{P}(x_i) \rightarrow (\pi_i, \mathcal{D}_i), \quad \pi_i = \langle s_{i1}, s_{i2}, \ldots, s_{im} \rangle, \quad \mathcal{D}_i = \langle d_{i1}, d_{i2}, \ldots, d_{im} \rangle.$$

Here, $\pi_i$ denotes the decomposition plan, a sequence of $m$ sub-questions, where each $s_{ij}$ is a natural language prompt targeting a specific sub-problem of the query $x_i$. The vector $\mathcal{D}_i = \langle d_{i1}, d_{i2}, \ldots, d_{im} \rangle$ contains corresponding complexity scores, with each $d_{ij} \in \mathbb{R}_{>0}$ reflecting the estimated complexity of solving $s_{ij}$ based on LLM confidence, problem structure, or other heuristics.

The decomposition plan $\pi_i$ is not unique or guaranteed to be optimal, but acts as a *soft scaffold*, a plausible high-level reasoning path as a prompt to guide the main LLM. The planning function $\mathcal{P}$ can be implemented via applying a decomposition prompt in a lightweight LLM (see Section H). The resulting complexity scores $d_{ij}$ reflect epistemic uncertainty and help estimate the computational effort required for each sub-question. These scores are then normalized into a weight vector $\mathbf{w}_i$:

$$w_{ij} = \frac{d_{ij}}{\sum_{k=1}^{m} d_{ik}}.$$

This normalized weight $w_{ij}$ represents the proportion of the total "complexity" that is attributed to the $j$-th sub-question. This weight vector then plays a key role in the budget allocation mechanism, determining how the total token budget $B_i$ is distributed across the individual sub-questions.

**Phase 2: Guided Reasoning.** After decomposing $x_i$ into sub-questions $\langle s_{i1}, \ldots, s_{im} \rangle$ and allocating token budgets $b_{i1}, \ldots, b_{im}$, the main reasoning LLM is guided by these sub-questions (see the prompt template in Section H). Each sub-question $s_{ij}$ is answered according to its allocated budget $b_{ij}$, yielding responses $a_{ij} = f_{\text{LLM}}(s_{ij}, b_{ij})$, where $f_{\text{LLM}}$ denotes the budget-constrained generation process. After all sub-questions are answered, a synthesis function $\mathcal{S}$ aggregates the responses, which answers the original query $x_i$: $y_i = S(a_{i1}, \ldots, a_{im})$.

### 4.2 BUDGET STEP: DECAY-BASED BUDGET ALLOCATION

While our Bayesian formulation offers an optimal allocation strategy based on sub-question-specific uncertainty parameters ($c_{ij}$ and $\beta_{ij}$), estimating these values reliably in practice is often infeasible. To bridge this gap, we introduce a family of *decay-based scheduling functions* that approximate uncertainty-aware budget allocation in a lightweight and practical manner.

Table 2: Decay-based scheduling strategies for token budget allocation.

| Strategy | Formula of $d_{ij}$ | Description |
|---|---|---|
| Non-decay | 1 | Equal priority for all sub-questions; budget follows $w_{ij}$. |
| Linear decay | $m - j$ | Decreases priority linearly with $j$; emphasizes early steps. |
| Polynomial decay | $(m - j)^p$ | Stronger emphasis on early steps; steeper with higher $p > 1$. |
| Exponential decay | $\gamma^j$ | Exponentially favors earlier sub-questions; controlled by $\gamma \in (0, 1)$. |
| Cosine annealing | $0.5 \left( 1 + \cos \left( \frac{\pi j}{m-1} \right) \right) + \epsilon$ | Smooth decay with mid-sequence flexibility; $\epsilon$ adds stability. |

These functions allocate more tokens to early sub-questions, based on the observation that epistemic uncertainty is typically highest at the start of reasoning—when foundational understanding and strategy formation occur. Early token usage yields greater uncertainty reduction, consistent with the power law behavior of epistemic uncertainty in Equation 1. In contrast, later steps are generally narrower in scope or more deterministic, and over-allocating tokens at these stages risks wasting inference effort, as additional computation cannot reduce the irreducible aleatoric uncertainty and yields diminishing returns in epistemic gain. Thus, decay functions offer a principled heuristic for prioritizing the budget where it is most valuable.

Given the normalized complexity weight vector $\mathbf{w}_i = \{w_{i1}, \ldots, w_{im}\}$ for a query $x_i$ and the total token budget $B_i$, we allocate tokens using

$$b_{ij} = \left\lfloor \frac{w_{ij} \cdot \rho_{ij}}{\sum_{k=1}^m w_{ik} \cdot \rho_{ik}} \cdot B_i \right\rfloor, \tag{7}$$

where $w_{ij}$ reflects the LLM-estimated relative complexity of sub-question $j$, and $\rho_{ij} = \texttt{schedule}(j, m)$ encodes a positional prior that can emphasize earlier reasoning stages. The allocation thus combines content-driven complexity weights with a structural decay prior, preserving both sub-question difficulty and stage-dependent uncertainty considerations.

**Experimental Scheduling Strategy.** We explore several decay strategies (Table 2), each encoding a distinct prioritization schema over sub-question positions. Each strategy offers a flexible way to encode task-specific preferences. For instance, polynomial decay aggressively front-loads the budget, which may be beneficial in highly ambiguous tasks. Exponential decay offers a more balanced approach for problems with both early and mid-sequence challenges. Ultimately, these decay functions serve as practical surrogates to our Bayesian-optimal allocation by heuristically targeting the most epistemically impactful stages of reasoning.

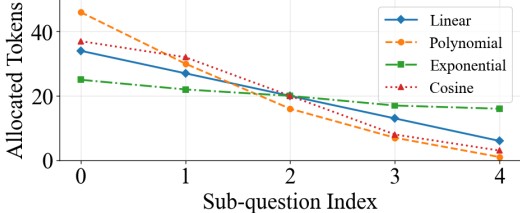

Figure 2: Visualization of decay functions $\rho$. We take $B = 100$, $p = 2$, $\gamma = 0.9$, and 5 sub-questions with the same complexity as an example.

Figure 2 shows that different decay strategies yield distinct allocation patterns even under uniform complexity, with polynomial decay and cosine annealing favoring early steps, linear decay offering gradual decline, and exponential decay providing a balanced distribution—demonstrating that decay-based scheduling flexibly adapts token emphasis to match the structure of reasoning tasks.

## 5 EXPERIMENTS

We conduct extensive experiments across three types of reasoning-intensive downstream tasks to evaluate the effectiveness and efficiency of PLAN-AND-BUDGET. We assess performance in terms of raw accuracy and compute-aware reasoning efficiency using our proposed $\mathcal{E}^3$ metric. In particular, we aim to answer the following questions: **Q1**: *Does Plan-and-Budget improve reasoning efficiency without sacrificing accuracy*, compared to the baseline of using no planning (Vanilla) or applying a fixed budget (Global Budget)? **Q2**: *How does local, uncertainty-aware budgeting perform across models, datasets, and task types*, relative to uniform or global strategies? **Q3**: *Which scheduling strategies yield the best efficiency–accuracy tradeoff?*

Table 3: Experiment results across different reasoning models on MATH-500. Acc denotes accuracy.

| Models → | DeepSeek-R1-Distill-Qwen-32B | | | QwQ-32B | | | DeepSeek-R1-Distill-Llama-70B | | | o4-mini | | |
|---|---|---|---|---|---|---|---|---|---|---|---|---|
| Methods↓ | Acc (%)↑ | Avg. Tok.↓ | $\mathcal{E}^3$↑ | Acc (%)↑ | Avg. Tok.↓ | $\mathcal{E}^3$↑ | Acc (%)↑ | Avg. Tok.↓ | $\mathcal{E}^3$↑ | Acc (%)↑ | Avg. Tok.↓ | $\mathcal{E}^3$↑ |
| **Direct** Vanilla | $89.76_{\pm0.26}$ | $2105.12_{\pm31.94}$ | 3.83 | $84.88_{\pm1.18}$ | $3523.72_{\pm97.42}$ | 2.04 | $90.44_{\pm0.61}$ | $2286.63_{\pm26.42}$ | 3.58 | $\mathbf{93.16_{\pm0.89}}$ | $711.20_{\pm8.31}$ | 12.20 |
| **Direct** Global Budget | $89.60_{\pm0.88}$ | $1526.15_{\pm10.09}$ | 5.26 | $\mathbf{90.56_{\pm0.33}}$ | $2565.18_{\pm37.10}$ | 3.20 | $90.80_{\pm0.62}$ | $1810.83_{\pm51.64}$ | 4.55 | $91.84_{\pm0.48}$ | $636.41_{\pm8.14}$ | 13.25 |
| **Planned** Vanilla | $\mathbf{91.04_{\pm0.62}}$ | $1883.73_{\pm63.82}$ | 4.40 | $85.30_{\pm1.56}$ | $3309.69_{\pm18.06}$ | 2.20 | $92.12_{\pm1.16}$ | $2022.38_{\pm28.74}$ | 4.20 | $91.88_{\pm1.36}$ | $539.36_{\pm18.94}$ | 15.65 |
| **Planned** Global Budget | $91.24_{\pm1.34}$ | $1552.62_{\pm29.93}$ | 5.36 | $88.20_{\pm1.17}$ | $2671.60_{\pm15.02}$ | 2.91 | $92.56_{\pm0.71}$ | $1661.24_{\pm34.43}$ | 5.16 | $91.84_{\pm0.75}$ | $586.18_{\pm6.50}$ | 14.39 |
| **Plan-and-Budget** + Uniform | $90.16_{\pm0.74}$ | $1440.70_{\pm47.55}$ | 5.64 | $88.68_{\pm0.58}$ | $2397.16_{\pm23.01}$ | 3.28 | $92.28_{\pm0.41}$ | $1575.04_{\pm29.68}$ | 5.41 | $91.36_{\pm0.85}$ | $525.53_{\pm18.88}$ | 15.88 |
| **Plan-and-Budget** + Weighted | $90.48_{\pm0.46}$ | $1485.99_{\pm45.63}$ | 5.51 | $87.45_{\pm0.66}$ | $2479.46_{\pm39.21}$ | 3.08 | $92.64_{\pm0.68}$ | $1557.64_{\pm47.71}$ | 5.51 | $91.64_{\pm1.21}$ | $538.22_{\pm5.30}$ | 15.60 |
| **Plan-and-Budget** + Linear | $90.04_{\pm0.46}$ | $\mathbf{1336.27_{\pm31.18}}$ | **6.07** | $88.13_{\pm0.90}$ | $2346.35_{\pm25.33}$ | 3.31 | $92.32_{\pm0.88}$ | $1529.98_{\pm45.35}$ | 5.57 | $90.56_{\pm0.73}$ | $534.45_{\pm7.64}$ | 15.34 |
| **Plan-and-Budget** + Exponential | $90.80_{\pm0.68}$ | $1389.75_{\pm61.06}$ | 5.93 | $87.90_{\pm1.27}$ | $2320.04_{\pm72.33}$ | 3.33 | $93.04_{\pm0.22}$ | $\mathbf{1469.29_{\pm73.77}}$ | **5.89** | $90.88_{\pm0.36}$ | $525.51_{\pm11.70}$ | 15.72 |
| **Plan-and-Budget** + Polynomial | $90.04_{\pm0.26}$ | $1371.59_{\pm21.75}$ | 5.91 | $88.27_{\pm0.99}$ | $2346.94_{\pm17.73}$ | 3.32 | $91.92_{\pm1.15}$ | $1514.43_{\pm47.94}$ | 5.58 | $90.36_{\pm0.83}$ | $525.00_{\pm9.15}$ | 15.55 |
| **Plan-and-Budget** + Cosine | $89.88_{\pm1.72}$ | $1365.51_{\pm44.92}$ | 5.92 | $88.60_{\pm0.28}$ | $\mathbf{2306.83_{\pm24.11}}$ | **3.40** | $\mathbf{92.88_{\pm0.46}}$ | $1487.83_{\pm61.78}$ | 5.80 | $91.32_{\pm0.94}$ | $\mathbf{522.89_{\pm10.01}}$ | **15.95** |

## 5.1 EXPERIMENT SETUP

**Datasets.** We evaluate PLAN-AND-BUDGET on three representative benchmarks (dataset statistics are shown in Table 15): (1) **MATH-500** (Lightman et al., 2024), a 500 math problem dataset requiring multi-step symbolic reasoning, evaluated by accuracy; (2) **NaturalInstructions** (Wang et al., 2022), a diverse instruction-following benchmark, evaluated using the ROUGE score; and (3) **TravelPlanner** (Xie et al., 2024), a challenging agentic planning task evaluated by a hard constraint pass rate in a tool-free setting. This benchmark reflects the difficulty of long-horizon, constraint-satisfying reasoning; prior work reports that even strong proprietary models achieve limited performance, highlighting the task's inherent complexity.

**Models.** We evaluate our methods on four state-of-the-art, publicly-available reasoning-tuned LLMs: DeepSeek-R1-Distill-Qwen-32B (**DS-Qwen-32B**) (Guo et al., 2025), **QwQ-32B** (Team, 2025), DeepSeek-R1-Distill-LLaMA-70B (**DS-LLaMA-70B**) (Guo et al., 2025), and OpenAI **o4-mini** (OpenAI, 2025). These models provide strong reasoning performance while remaining publicly available and reproducible. For planning and budgeting, we use a lightweight non-reasoning LLM, LLaMA-3.1-8B-Instruct (Grattafiori et al., 2024). To ensure that it does not inadvertently contribute to final answer quality, we evaluate its standalone performance on the three benchmarks and find that it underperforms specialized models: $48.76_{\pm0.74}$ on MATH-500, $21.72_{\pm0.98}$ on NaturalInstructions, and $2.91_{\pm0.28}$ on TravelPlanner. These results confirm that improvements arise from structured planning and adaptive budgeting rather than the planner model's intrinsic reasoning capability.

**Baselines.** We compare our proposed framework against several baselines: (1) **Vanilla.** The query is given to the LLM without planning or token constraints; (2) **Global Budget.** Same as Vanilla but with a token limit prompt (e.g., "use fewer than $B_i$ tokens"); (3) **Planned Vanilla / Global Budget.** Same as above, but with the original query and its decomposed sub-questions provided; and (4) **PLAN-AND-BUDGET.** Our method, which decomposes the query, estimates sub-question complexity, computes a local budget allocation, and enforces these allocations through step-specific token constraints. We explore several scheduling strategies for local allocation: (a) **Uniform**, equal tokens per sub-question; (b) **Weighted**, proportional to the estimated difficulty; and (c) **Linear, Polynomial, Exponential, Cosine**, weighted by difficulty with additional decay (we use $p = 2$ and $\gamma = 0.9$). A hard cutoff of 8192 tokens is applied to prevent runaway generations. We report the average and standard deviation over 5 runs for all models and baselines.

**Evaluation Metrics.** We report the following metrics: (1) **Score (%)**, the original evaluation metric used in each dataset; (2) **Avg. Tokens**, the average number of all billed completion tokens per query, including planning, reasoning and output tokens (for open-source models, tokens before `</think>` and final outputs; for o4-mini, the sum of reasoning and output tokens as reported in the OpenAI documentation (OpenAI, 2025)); and (3) $\mathcal{E}^3$, which captures the balance between correctness and computational cost.

## 5.2 COMPARATIVE RESULTS

We now address the questions introduced earlier by analyzing results across datasets and models.

Tables 3–5 summarize our main findings. Across all datasets and model scales, PLAN-AND-BUDGET consistently outperforms both the Vanilla and Global Budget baselines, achieving up to **193.8% improvement in** $\mathcal{E}^3$, while maintaining comparable or even higher accuracy. Figure 3 shows answer pass rates of QwQ-32B on TravelPlanner, grouped by difficulty. While global budgets reduce token usage,

Table 4: Experiment results across different reasoning models on NaturalInstructions.

| Models → | DeepSeek-R1-Distill-Qwen-32B | | | QwQ-32B | | | DeepSeek-R1-Distill-Llama-70B | | | o4-mini | | |
|---|---|---|---|---|---|---|---|---|---|---|---|---|
| Methods↓ | ROUGE (%)↑ | Avg. Tokens↓ | $\mathcal{E}^3$↑ | ROUGE (%)↑ | Avg. Tokens↓ | $\mathcal{E}^3$↑ | ROUGE (%)↑ | Avg. Tokens↓ | $\mathcal{E}^3$↑ | ROUGE (%)↑ | Avg. Tokens↓ | $\mathcal{E}^3$↑ |
| **Direct** Vanilla | **43.47**$_{\pm 0.52}$ | 968.17$_{\pm 44.78}$ | 1.95 | 43.16$_{\pm 1.12}$ | 1818.34$_{\pm 24.99}$ | 1.02 | 43.13$_{\pm 0.76}$ | 894.46$_{\pm 50.69}$ | 2.08 | **47.24**$_{\pm 0.31}$ | 460.99$_{\pm 11.31}$ | 4.84 |
| **Direct** Global Budget | 42.81$_{\pm 0.39}$ | 787.25$_{\pm 58.17}$ | 2.33 | **44.77**$_{\pm 0.73}$ | 1360.49$_{\pm 101.64}$ | 1.47 | **43.80**$_{\pm 1.28}$ | 772.98$_{\pm 47.44}$ | 2.48 | 45.39$_{\pm 1.27}$ | 422.20$_{\pm 56.78}$ | 4.88 |
| **Planned** Vanilla | 42.48$_{\pm 0.67}$ | 860.85$_{\pm 49.58}$ | 2.10 | 44.24$_{\pm 0.67}$ | 1426.74$_{\pm 52.92}$ | 1.37 | 43.40$_{\pm 0.18}$ | 821.27$_{\pm 21.85}$ | 2.29 | 43.78$_{\pm 1.47}$ | **344.99**$_{\pm 14.44}$ | 5.56 |
| **Planned** Global Budget | 42.50$_{\pm 0.36}$ | 717.98$_{\pm 36.28}$ | 2.52 | 45.13$_{\pm 0.56}$ | 1265.78$_{\pm 33.23}$ | 1.61 | 42.48$_{\pm 0.33}$ | 691.79$_{\pm 12.18}$ | 2.61 | 43.78$_{\pm 0.96}$ | 358.84$_{\pm 14.44}$ | 5.34 |
| **Plan-and-Budget** + Uniform | 41.03$_{\pm 0.55}$ | 644.87$_{\pm 46.34}$ | 2.61 | 44.47$_{\pm 0.35}$ | 996.91$_{\pm 31.31}$ | 1.98 | 43.06$_{\pm 0.33}$ | 665.94$_{\pm 47.22}$ | 2.78 | 44.08$_{\pm 0.81}$ | 348.74$_{\pm 8.13}$ | **5.57** |
| + Weighted | 41.29$_{\pm 0.50}$ | 663.94$_{\pm 27.29}$ | 2.57 | 44.40$_{\pm 0.61}$ | 1025.02$_{\pm 24.91}$ | 1.92 | 43.05$_{\pm 0.39}$ | 626.37$_{\pm 19.46}$ | **2.96** | 43.72$_{\pm 1.00}$ | 371.85$_{\pm 9.53}$ | 5.14 |
| + Linear | 41.56$_{\pm 0.50}$ | 633.79$_{\pm 34.17}$ | 2.73 | 44.22$_{\pm 0.66}$ | 1003.24$_{\pm 26.23}$ | 1.95 | 42.05$_{\pm 0.99}$ | **613.05**$_{\pm 33.68}$ | 2.88 | 44.21$_{\pm 0.44}$ | 363.65$_{\pm 13.70}$ | 5.37 |
| + Exponential | 41.44$_{\pm 0.50}$ | 650.19$_{\pm 31.35}$ | 2.64 | 43.99$_{\pm 0.22}$ | 1026.89$_{\pm 8.51}$ | 1.88 | 42.73$_{\pm 0.24}$ | 622.72$_{\pm 33.58}$ | 2.93 | 43.68$_{\pm 1.06}$ | 364.86$_{\pm 10.81}$ | 5.23 |
| + Polynomial | 41.44$_{\pm 0.78}$ | **600.04**$_{\pm 40.52}$ | **2.86** | 44.66$_{\pm 0.68}$ | **995.95**$_{\pm 14.43}$ | **2.00** | 43.19$_{\pm 0.44}$ | 641.62$_{\pm 32.22}$ | 2.91 | 44.63$_{\pm 1.04}$ | 363.16$_{\pm 11.71}$ | 5.48 |
| + Cosine | 41.43$_{\pm 1.01}$ | 628.20$_{\pm 36.63}$ | 2.73 | 44.53$_{\pm 0.54}$ | 1000.64$_{\pm 17.85}$ | 1.98 | 42.83$_{\pm 0.63}$ | 657.93$_{\pm 59.06}$ | 2.79 | 44.36$_{\pm 1.06}$ | 363.05$_{\pm 16.72}$ | 5.42 |

Table 5: Experiment results on TravelPlanner. Rate denotes the hard constraint pass rate.

| Models → | DeepSeek-R1-Distill-Qwen-32B | | | QwQ-32B | | | DeepSeek-R1-Distill-Llama-70B | | | o4-mini | | |
|---|---|---|---|---|---|---|---|---|---|---|---|---|
| Methods↓ | Rate (%)↑ | Avg. Tokens↓ | $\mathcal{E}^3$↑ | Rate (%)↑ | Avg. Tokens↓ | $\mathcal{E}^3$↑ | Rate (%)↑ | Avg. Tokens↓ | $\mathcal{E}^3$↑ | Rate (%)↑ | Avg. Tokens↓ | $\mathcal{E}^3$↑ |
| **Direct** Vanilla | 14.33$_{\pm 2.17}$ | 1430.14$_{\pm 43.73}$ | 0.14 | 34.89$_{\pm 3.20}$ | 3432.33$_{\pm 78.66}$ | 0.35 | 26.22$_{\pm 1.82}$ | 1361.37$_{\pm 47.93}$ | 0.50 | 11.58$_{\pm 2.15}$ | 1559.65$_{\pm 8.84}$ | 0.086 |
| **Direct** Global Budget | 13.78$_{\pm 1.20}$ | 1158.81$_{\pm 20.23}$ | 0.16 | 30.78$_{\pm 2.06}$ | 2530.04$_{\pm 40.87}$ | 0.37 | 24.33$_{\pm 2.30}$ | 1215.29$_{\pm 35.05}$ | 0.49 | 8.33$_{\pm 1.71}$ | 1248.53$_{\pm 26.97}$ | 0.056 |
| **Planned** Vanilla | 20.22$_{\pm 1.01}$ | 1343.67$_{\pm 62.44}$ | 0.30 | **37.22**$_{\pm 1.80}$ | 3669.88$_{\pm 42.09}$ | 0.38 | 30.67$_{\pm 2.17}$ | 1464.50$_{\pm 65.40}$ | 0.64 | **12.20**$_{\pm 2.47}$ | 1640.46$_{\pm 95.33}$ | 0.091 |
| **Planned** Global Budget | 22.56$_{\pm 2.41}$ | 1241.19$_{\pm 54.66}$ | 0.41 | 35.22$_{\pm 4.85}$ | 3199.58$_{\pm 63.14}$ | 0.39 | 30.67$_{\pm 1.73}$ | 1220.41$_{\pm 32.22}$ | 0.77 | 7.19$_{\pm 2.43}$ | 1392.11$_{\pm 31.05}$ | 0.037 |
| **Plan-and-Budget** + Uniform | 20.67$_{\pm 1.20}$ | 1227.99$_{\pm 68.55}$ | 0.35 | 36.00$_{\pm 2.79}$ | 2854.24$_{\pm 44.87}$ | 0.45 | 31.56$_{\pm 2.20}$ | 1232.98$_{\pm 34.16}$ | 0.81 | 11.00$_{\pm 1.62}$ | 1345.32$_{\pm 58.88}$ | 0.090 |
| + Weighted | **23.33**$_{\pm 1.11}$ | 1222.09$_{\pm 40.69}$ | 0.45 | 33.89$_{\pm 2.22}$ | 2842.74$_{\pm 77.68}$ | 0.40 | 29.67$_{\pm 3.01}$ | 1197.32$_{\pm 10.78}$ | 0.74 | 10.91$_{\pm 3.01}$ | 1353.67$_{\pm 37.64}$ | 0.088 |
| + Linear | 19.56$_{\pm 2.47}$ | **1136.18**$_{\pm 54.92}$ | 0.34 | 34.55$_{\pm 2.65}$ | 2671.70$_{\pm 67.97}$ | 0.45 | 31.67$_{\pm 2.32}$ | 1162.24$_{\pm 43.31}$ | 0.86 | 11.66$_{\pm 1.96}$ | 1306.54$_{\pm 55.05}$ | 0.103 |
| + Exponential | 21.44$_{\pm 2.98}$ | 1156.64$_{\pm 30.52}$ | 0.40 | 35.44$_{\pm 2.06}$ | 2724.23$_{\pm 41.87}$ | 0.46 | 32.00$_{\pm 2.14}$ | 1187.85$_{\pm 36.57}$ | 0.86 | 9.91$_{\pm 1.96}$ | 1307.87$_{\pm 40.83}$ | 0.075 |
| + Polynomial | 23.11$_{\pm 2.14}$ | 1148.53$_{\pm 37.33}$ | **0.47** | 35.00$_{\pm 3.35}$ | 2511.35$_{\pm 84.18}$ | 0.49 | **32.67**$_{\pm 2.06}$ | 1148.14$_{\pm 59.00}$ | **0.93** | 11.49$_{\pm 1.31}$ | 1266.11$_{\pm 28.48}$ | **0.104** |
| + Cosine | 20.22$_{\pm 2.34}$ | 1140.79$_{\pm 6.68}$ | 0.36 | 36.18$_{\pm 3.00}$ | **2496.46**$_{\pm 40.10}$ | **0.52** | 31.67$_{\pm 2.22}$ | 1173.96$_{\pm 44.22}$ | 0.85 | 9.79$_{\pm 1.57}$ | 1252.06$_{\pm 80.85}$ | 0.077 |

they also degrade pass rates across all levels. In contrast, PLAN-AND-BUDGET achieves both higher pass rates and lower token usage, especially on harder queries, highlighting its ability to scale reasoning adaptively with problem complexity.

On MATH-500, our method improves $\mathcal{E}^3$ consistently by over 20%—for instance, from 4.55 → 5.89 (+29.4%) on DS-LLaMA-70B and from 13.25 → 15.95 (+20.3%) on o4-mini. Importantly, this is achieved without compromising the accuracy. While the Global Budget baseline reduces token usage, its gains are limited due to a lack of uncertainty-awareness. In particu-

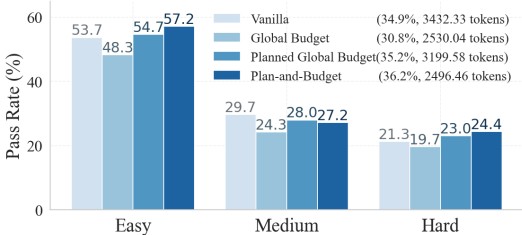

Figure 3: Answer pass rate (%) grouped by the question difficulty level, with legend showing the overall pass rate (%) and average token usage. The global budget limit hurts the pass rate on all levels, while our method not only achieves a higher pass rate but also enjoys lower token usage.

lar, we find that **planning alone** (Planned Global Budget) already increases efficiency by 2–13%, validating our first key principle: *reasoning should be structured*. This scaffolding greatly reduces speculative exploration. Moreover, $\mathcal{E}^3$ enables easy comparison across models—e.g., o4-mini consistently achieves the highest $\mathcal{E}^3$, despite having similar accuracy to other models, because it uses the fewest tokens. This underscores the importance of $\mathcal{E}^3$ as a practical efficiency metric.

**A1: We achieve substantial efficiency gains with comparable accuracy.** On NaturalInstructions, PLAN-AND-BUDGET improves $\mathcal{E}^3$ by 19.3–36.0%. For example, on QwQ-32B, it improves from 1.47 → 2.00 (+36%), and on o4-mini, from 4.88 → 5.57 (+14%). Although these tasks are more instruction-oriented, PLAN-AND-BUDGET remains beneficial. On TravelPlanner, the most open-ended and challenging benchmark, we observe the most dramatic gains: $\mathcal{E}^3$ improves from 0.16 → 0.47 (+193.8%) on DS-Qwen-32B, from 0.49 → 0.93 (+89.8%) on DS-LLaMA-70B, and 0.056 → 0.104 (+85.7%) on o4-mini. These results highlight that **the more complex task, the greater the benefit of structure and adaptivity.**

**A2: Local budgeting consistently improves efficiency.** While structured planning alone improves efficiency, adding local budgeting yields significant additional gains. We observe that on MATH-500, DS-LLaMA-70B improves $\mathcal{E}^3$ from 5.16 → 5.89 (+14.1%); on NaturalInstructions, QwQ-32B improves from 1.61 → 2.00 (+24.2%); and on TravelPlanner, QwQ-32B improves from 0.39 → 0.52 (+33.3%). These results confirm the importance of adapting the budget to the sub-question, rather than applying a global allocation.

**A3: Front-loaded scheduling performs best on complex tasks.** Among local budget schedulers, polynomial decay and cosine annealing consistently deliver the highest $\mathcal{E}^3$ on mathematical and long-form planning tasks. These strategies front-load computation, allocating more budget to early, uncertain steps where reasoning direction is established. This pattern is particularly effective on MATH-500 and TravelPlanner, where clarity at the beginning of the reasoning is crucial. In contrast, on NaturalInstructions, weighted or uniform schedules usually perform well, suggesting that smooth, evenly paced reasoning suffices for tasks with clearer structure and less ambiguity.

**A4: Bridging the gap between small and large models.** Our method is model-agnostic: it requires no retraining or fine-tuning, relying only on prompting and lightweight planning. We observe consistent improvements across model sizes, from small models like DS-Qwen-32B and QwQ-32B to large models like DeepSeek-R1-70B. An especially notable result comes from TravelPlanner, where a compact model (DS-Qwen-32B) originally achieved only $\mathcal{E}^3 = 0.16$, but reached $\mathcal{E}^3 = 0.47$ after applying PLAN-AND-BUDGET, on par with a larger model with no planning (DS-LLaMA-70B, $\mathcal{E}^3 = 0.50$). This demonstrates that planning and budgeting can serve as powerful inference-time equalizers, closing the gap between small and large models through better compute utilization.

## 5.3 ROBUSTNESS AND SENSITIVITY ANALYSIS

We conduct additional studies to further stress-test the framework (Section F).

**Early-termination baselines.** We compare against the recent early-termination baseline, Certaindex (Fu et al., 2024) across three representative LLMs. PLAN-AND-BUDGET consistently outperforms early termination in both efficiency and accuracy trade-offs (Table 7).

Table 6: The average latency (seconds) breakdown. The question decomposition + budgeting took 14.73 seconds on average.

| Models →
Methods ↓ | DeepSeek-R1-
Distill-Qwen-32B | QwQ-32B | DeepSeek-R1-
Distill-Llama-70B |
|---|---|---|---|
| Vanilla | 335.56 | 756.06 | 777.66 |
| Global Budget | 237.83 | 486.48 | 527.39 |
| PLAN-AND-BUDGET | **221.25**
(14.73 + 206.52) | **441.61**
(14.73 + 426.88) | **481.14**
(14.73 + 466.41) |

**Latency breakdown.** While planning introduces minor overheads, total wall-clock latency decreases due to reduced reasoning chains under adaptive allocation. The planning and budget allocation step contributes only 3-7% of the total end-to-end latency, while the adaptive budgeting reduces total latency by 38-44% compared to vanilla decoding, yielding a net efficiency gain (Table 6).

**Metric robustness.** Our primary evaluation metric $\mathcal{E}^3 = A^2/T$ more strongly penalizes accuracy degradation while rewarding computational efficiency. To ensure improvements are not metric-specific, we additionally evaluate using the conventional $A/T$ measure. Relative performance rankings remain stable across both metrics (Table 8–10), reinforcing that the observed gains are not artifacts of metric choice.

**Budget sensitivity.** When varying the prompted token budget, we observe consistent changes in realized token usage, demonstrating that budget prompts effectively regulate reasoning depth under different resource regimes (Table 11).

**Planner robustness.** Replacing the default planner with the stronger LLaMA 3.3-**70B** yields consistent improvements, indicating that gains are not tied to a specific planner (Table 12–14).

## 6 CONCLUSION

We propose PLAN-AND-BUDGET, a lightweight test-time framework that improves LLM reasoning efficiency by combining structured planning with uncertainty-aware token budgeting. Built on our Budget Allocation Model, PLAN-AND-BUDGET models reasoning as a sequence of sub-questions and adaptively allocates computation based on estimated difficulty. Experiments on three different reasoning tasks show that PLAN-AND-BUDGET achieves significant improvements in computational efficiency over strong baselines, without compromising accuracy. Although effective, our method currently requires an additional LLM call to generate the decomposition plan. In future work, we aim to fine-tune and develop a dedicated planner LLM to internalize the plan-and-budget strategy, enabling end-to-end, efficient reasoning within a single model.

## REPRODUCIBILITY STATEMENT

We have taken multiple steps to ensure reproducibility. Theoretical assumptions and derivations of the Budget Allocation Model (BAM) are detailed in Section 3, with complete proofs provided in Section C–E. Experimental setups, evaluation metrics, and dataset statistics are described in Section 5 and Section G, including licenses for all datasets and models. We provide an anonymized code repository (linked in the abstract) containing implementations of all baselines, our Plan-and-Budget framework, and scripts to reproduce every table and figure. Additional details, such as prompt templates and robustness and sensitivity analysis, are included in Section H and F. Together, these resources allow independent verification and extensions of both our theoretical and empirical findings.

## ETHICS STATEMENT

This work relies exclusively on publicly-available datasets (MATH-500, NaturalInstructions, and TravelPlanner) and open-source or API-accessible large language models (e.g., DeepSeek, QwQ, o4-mini). No human subjects, private, or sensitive data were used. The proposed method improves inference efficiency by reducing unnecessary computation, which can lower environmental and financial costs of deploying LLMs. However, as with any efficiency-focused technique, there is a risk of misuse in high-stakes applications (e.g., medical or legal decision-making) if efficiency is prioritized over accuracy. We mitigate this risk by explicitly emphasizing correctness in our $\mathcal{E}^3$ metric and by recommending careful, task-specific evaluation before real-world deployment. Our framework does not modify underlying model internals and thus inherits any limitations or biases present in the base models.

## ACKNOWLEDGEMENTS

We thank the anonymous reviewers for their constructive comments. This work is supported by the MIT-IBM AI Watson Lab, NSF awards #CCF-1845763, #CCF-2316235, #CCF-2403237, #IIS-2339989, and #2406439, Google Faculty Research Award, Google Research Scholar Award, DARPA under contract No. HR00112490370 and No. HR001124S0013, U.S. Department of Homeland Security under Grant Award No. 17STCIN00001-08-00, Amazon-Virginia Tech Initiative for Efficient and Robust Machine Learning, Amazon AWS, Google, Cisco, 4-VA, Commonwealth Cyber Initiative, National Surface Transportation Safety Center for Excellence, and Virginia Tech. The views and conclusions are those of the authors and should not be interpreted as representing the official policies of the funding agencies or the government.

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

## A    BROADER IMPACTS

Our work proposes a lightweight test-time framework that improves LLM reasoning efficiency through structured planning and uncertainty-aware computation. This has potential positive societal impacts by reducing computational costs, improving energy efficiency, and making advanced LLM capabilities more accessible, particularly in resource-constrained settings. By narrowing the performance gap between small and large models, our method may also promote more equitable access to language technologies.

However, as with any LLM inference technique, risks remain if deployed without careful oversight. More efficient reasoning pipelines could accelerate LLM integration into high-stakes applications (e.g., legal or medical decision-making) where accuracy, fairness, and robustness are critical. Our method does not modify model internals and inherits any limitations or biases present in the base models. Mitigation strategies include model-level auditing, task-specific evaluation, and responsible deployment practices.

## B    LLM USAGE

Large language models (LLMs) were employed in a limited and transparent manner during the preparation of this manuscript. Specifically, LLMs were used to assist with linguistic refinement, style adjustments, and minor text editing to improve clarity and readability. They were not involved in formulating the research questions, designing the theoretical framework, conducting experiments, or interpreting results. All scientific contributions, including conceptual development, methodology, analyses, and conclusions, are the sole responsibility of the authors.

## C    PROOF OF UNCERTAINTY DECOMPOSITION FOR LLMS

Let $\theta \in \mathbb{R}^d$ denote the parameters of an LLM (e.g., transformer weights), and $D = \{(x_i, y_i)\}_{i=1}^{N}$ represent the training dataset. The predictive distribution for a new test-time input $x^*$ and output $y^*$ is obtained by marginalizing over the posterior $p(\theta|D)$:

$$p(y^*|x^*, D) = \int p(y^*|x^*, \theta)p(\theta|D)\, d\theta,$$

and is often approximated via Monte Carlo sampling:

$$p(y^*|x^*, D) \approx \frac{1}{M} \sum_{m=1}^{M} p(y^*|x^*, \theta_m), \quad \theta_m \sim p(\theta|D).$$

We define the total predictive uncertainty as the Shannon entropy of this marginal predictive distribution:

$$\mathcal{U}(x^*) = \mathcal{H}\left[p(y^*|x^*, D)\right] = \mathcal{H}\left[\int p(y^*|x^*, \theta)p(\theta|D)d\theta\right].$$

To derive the decomposition, we apply the *law of total entropy*, which relates the entropy of the predictive distribution $p(y^*|x^*, D)$ to the expected entropy of the conditionals and the mutual information:

$$\mathcal{H}[y^*|x^*, D] = \mathbb{E}_{p(\theta|D)}[\mathcal{H}[y^*|x^*, \theta]] + \mathcal{I}(y^*; \theta|x^*, D).$$

**Step-by-step Derivation:**

Let $p(y^*|x^*, \theta)$ be the conditional predictive distribution and $p(y^*|x^*, D)$ be the marginal (Bayesian averaged) predictive distribution.

The total predictive uncertainty is:

$$\mathcal{U}(x^*) = \mathcal{H}\left[p(y^*|x^*, D)\right] = -\sum_{y^*} p(y^*|x^*, D) \log p(y^*|x^*, D).$$

Define aleatoric uncertainty as the expected conditional entropy:

$$\mathcal{U}_{\text{aleatoric}}(x^*) = \mathbb{E}_{p(\theta|D)}[\mathcal{H}[p(y^*|x^*, \theta)]] = \int p(\theta|D) \left( -\sum_{y^*} p(y^*|x^*, \theta) \log p(y^*|x^*, \theta) \right) d\theta.$$

Then define epistemic uncertainty as the mutual information:

$$\mathcal{U}_{\text{epistemic}}(x^*) = \mathcal{I}(y^*; \theta|x^*, D) = \mathcal{H}[y^*|x^*, D] - \mathbb{E}_{p(\theta|D)}[\mathcal{H}[y^*|x^*, \theta]].$$

Combining the above, we obtain:

$$\mathcal{U}(x^*) = \mathcal{U}_{\text{aleatoric}}(x^*) + \mathcal{U}_{\text{epistemic}}(x^*).$$

**Interpretation:**

- $\mathcal{U}_{\text{aleatoric}}(x^*)$: Irreducible uncertainty present in each individual model prediction, even if $\theta$ were known.
- $\mathcal{U}_{\text{epistemic}}(x^*)$: Captures model uncertainty due to limited data, reflected in disagreement across posterior samples.

In practice, following Hüllermeier & Waegeman (2021), we approximate this decomposition using Monte Carlo estimation. Drawing $M$ samples $\theta_1, \ldots, \theta_M$ from $p(\theta|D)$, we compute:

$$\mathcal{U}(x^*) \approx \mathcal{H}\left[\frac{1}{M} \sum_{m=1}^{M} p(y^*|x^*, \theta_m)\right],$$

$$\mathcal{U}_{\text{aleatoric}}(x^*) \approx \frac{1}{M} \sum_{m=1}^{M} \mathcal{H}\left[p(y^*|x^*, \theta_m)\right],$$

$$\mathcal{U}_{\text{epistemic}}(x^*) \approx \mathcal{U}(x^*) - \mathcal{U}_{\text{aleatoric}}(x^*).$$

Thus, the uncertainty decomposition holds in both exact Bayesian inference and its Monte Carlo approximation, validating its use in practical LLM reasoning pipelines. In the context of our PLAN-AND-BUDGET framework, we utilize this decomposition as a theoretical lens to explain why structured budgeting works. We do not perform the computationally expensive Monte Carlo sampling described above during inference; rather, we rely on the deterministic approximation that the model's greedy or most-likely generation path serves as a sufficient proxy for the underlying uncertainty distribution.

## D  PROOF OF LAGRANGE OPTIMALITY

We aim to maximize the total utility:

$$\mathcal{R}_{\text{total}} = \sum_{j=1}^{m} r(s_{ij} \mid b_{ij}) = \sum_{j=1}^{m} \alpha \left( 1 - \frac{c_{ij}}{b_{ij}^{\beta_{ij}}} - \mathcal{U}_{\text{aleatoric}}(s_{ij}) \right). \tag{8}$$

Since $\alpha$ and $\mathcal{U}_{\text{aleatoric}}(s_{ij})$ are constants with respect to $b_{ij}$, maximizing the total utility is equivalent to minimizing the following:

$$\sum_{j=1}^{m} \frac{c_{ij}}{b_{ij}^{\beta_{ij}}} \quad \text{subject to} \quad \sum_{j=1}^{m} b_{ij} = B_i. \tag{9}$$

**Step 1: Form the Lagrangian.**
We define the Lagrangian:

$$\mathcal{L}(\{b_{ij}\}, \lambda) = \sum_{j=1}^{m} \frac{c_{ij}}{b_{ij}^{\beta_{ij}}} + \lambda \left( \sum_{j=1}^{m} b_{ij} - B_i \right). \tag{10}$$

Taking the partial derivative with respect to $b_{ij}$ and setting it to zero, we obtain:

$$\frac{\partial \mathcal{L}}{\partial b_{ij}} = -c_{ij}\beta_{ij}b_{ij}^{-(\beta_{ij}+1)} + \lambda = 0 \quad \Rightarrow \quad \lambda = c_{ij}\beta_{ij}b_{ij}^{-(\beta_{ij}+1)}. \tag{11}$$

Solving for $b_{ij}$ gives:

$$b_{ij}^{\beta_{ij}+1} = \frac{c_{ij}\beta_{ij}}{\lambda} \quad \Rightarrow \quad b_{ij} = \left(\frac{c_{ij}\beta_{ij}}{\lambda}\right)^{\frac{1}{\beta_{ij}+1}}. \tag{12}$$

**Step 2: Apply the budget constraint.**
Substitute into the constraint $\sum_j b_{ij} = B_i$:

$$\sum_{j=1}^{m} \left(\frac{c_{ij}\beta_{ij}}{\lambda}\right)^{\frac{1}{\beta_{ij}+1}} = B_i. \tag{13}$$

Let

$$A_j := (c_{ij}\beta_{ij})^{\frac{1}{\beta_{ij}+1}}, \quad \text{so that} \quad b_{ij} = \lambda^{-1/(\beta_{ij}+1)}A_j.$$

Then, the constraint becomes:

$$\sum_{j=1}^{m} \lambda^{-1/(\beta_{ij}+1)}A_j = B_i. \tag{14}$$

This expression yields a closed-form solution for $\lambda$ only under the assumption of homogeneous difficulty, where $\beta_{ij} = \beta$ for all sub-questions $j$. Under this assumption, $\lambda^{-1/(\beta+1)}$ becomes a common factor across the summation, allowing us to solve for the optimal budget allocation $b_{ij}$ analytically. In the more general case of heterogeneous difficulty, $\lambda$ must be determined numerically to satisfy the budget constraint $\sum b_{ij} = B_i$. In this case:

$$b_{ij} = \left(\frac{c_{ij}\beta}{\lambda}\right)^{\frac{1}{\beta+1}} \quad \Rightarrow \quad \sum_j \left(\frac{c_{ij}\beta}{\lambda}\right)^{\frac{1}{\beta+1}} = B_i \quad \Rightarrow \quad \lambda = \left(\frac{\sum_j(c_{ij}\beta)^{\frac{1}{\beta+1}}}{B}\right)^{\beta+1}. \tag{15}$$

Substituting back yields:

$$b_{ij}^* = B_i \cdot \frac{(c_{ij}\beta)^{\frac{1}{\beta+1}}}{\sum_k(c_{ik}\beta)^{\frac{1}{\beta+1}}}. \tag{16}$$

In the general case of heterogeneous $\beta_{ij}$, the normalized form can still be written as:

$$b_{ij}^* = B_i \cdot \frac{(c_{ij}\beta_{ij})^{\frac{1}{\beta_{ij}+1}}}{\sum_k(c_{ik}\beta_{ik})^{\frac{1}{\beta_{ik}+1}}}, \tag{17}$$

which satisfies the budget constraint $\sum_j b_{ij} = B_i$, thus completing the proof.

## E    ANALYSIS OF THE RELATIONSHIP BETWEEN $b_{ij}$ AND $\beta_{ij}$

We examine the behavior of the allocation function in Equation 6:

$$b_{ij} = B_i \cdot \frac{(c_{ij}\beta_{ij})^{\frac{1}{\beta_{ij}+1}}}{\sum_k(c_{ik}\beta_{ik})^{\frac{1}{\beta_{ik}+1}}}. \tag{18}$$

To analyze the relationship between $b_{ij}$ and $\beta_{ij}$, we focus on the numerator:

$$f(\beta) := (\beta_{ij}c_{ij})^{\frac{1}{\beta_{ij}+1}} = \exp\left(\frac{\log(\beta_{ij}c_{ij})}{\beta_{ij}+1}\right). \tag{19}$$

Table 7: Experiment results comparing with the early termination baseline, **Certaindex** Fu et al. (2024), on MATH-500. Acc denotes accuracy.

| Models → | DeepSeek-R1-Distill-Qwen-32B | | | QwQ-32B | | | DeepSeek-R1-Distill-Llama-70B | | |
|---|---|---|---|---|---|---|---|---|---|
| Methods↓ | Acc (%) ↑ | Avg. Tok.↓ | $\mathcal{E}^3$ ↑ | Acc (%) ↑ | Avg. Tok.↓ | $\mathcal{E}^3$ ↑ | Acc (%) ↑ | Avg. Tok.↓ | $\mathcal{E}^3$ ↑ |
| Vanilla | $89.76_{\pm 0.26}$ | $2105.12_{\pm 31.94}$ | 3.83 | $84.88_{\pm 1.18}$ | $3523.72_{\pm 97.42}$ | 2.04 | $90.44_{\pm 0.61}$ | $2286.63_{\pm 26.42}$ | 3.58 |
| Global Budget | $89.60_{\pm 0.88}$ | $1526.15_{\pm 10.09}$ | 5.26 | $\mathbf{90.56}_{\pm 0.33}$ | $2565.18_{\pm 37.10}$ | 3.20 | $90.80_{\pm 0.62}$ | $1810.83_{\pm 51.64}$ | 4.55 |
| PLAN-AND-BUDGET | $\mathbf{90.04}_{\pm 0.46}$ | $\mathbf{1336.27}_{\pm 31.18}$ | **6.07** | $88.60_{\pm 0.28}$ | $\mathbf{2306.83}_{\pm 24.11}$ | **3.40** | $\mathbf{93.04}_{\pm 0.22}$ | $\mathbf{1469.29}_{\pm 73.77}$ | **5.89** |
| Certaindex | $81.44_{\pm 1.38}$ | $1985.57_{\pm 46.12}$ | 3.34 | $86.12_{\pm 0.73}$ | $5141.72_{\pm 82.95}$ | 1.44 | $81.96_{\pm 1.24}$ | $2200.68_{\pm 79.88}$ | 3.05 |

Let us define:

$$g(\beta) := \frac{\log(\beta c)}{\beta + 1}, \quad \text{so that} \quad f(\beta) = e^{g(\beta)}. \tag{20}$$

We now study the behavior of $f(\beta)$ through the derivative of $g(\beta)$:

$$g'(\beta) = \frac{1}{\beta + 1} \cdot \frac{1}{\beta} - \frac{\log(\beta c)}{(\beta + 1)^2} \tag{21}$$

$$= \frac{1}{\beta(\beta + 1)} - \frac{\log(\beta c)}{(\beta + 1)^2}. \tag{22}$$

The sign of $g'(\beta)$ depends on $\beta$, and it is not monotonic. The function $g(\beta)$ increases initially, reaches a maximum, and then decreases. Consequently, $g(\beta)$ is **unimodel**, and since $f(\beta) = e^{g(\beta)}$, $f(\beta)$ is also unimodal.

# F  ADDITIONAL RESULTS

## F.1  ADDITIONAL BASELINES: EARLY TERMINATION

We compare PLAN-AND-BUDGET against a representative early-termination strategy (Certaindex from Fu et al. (2024)), which dynamically stops generation when confidence exceeds a predefined threshold. Unlike PLAN-AND-BUDGET, which redistributes budget structurally across reasoning stages, early termination reduces computation by truncating later tokens without modifying the allocation structure.

As shown in Table 7, PLAN-AND-BUDGET consistently achieves superior efficiency–accuracy trade-offs across three LLMs. In particular, while Certaindex reduces token usage, it often sacrifices accuracy due to premature stopping (often even worse than Vanilla). In contrast, PLAN-AND-BUDGET preserves reasoning depth in high-uncertainty stages while eliminating redundant computation in later stages, leading to more balanced improvements.

## F.2  LATENCY BREAKDOWN

We analyze wall-clock latency by decomposing runtime into planning overhead and generation time. As shown in Table 6, while planning introduces modest overhead, total latency decreases due to shorter reasoning chains under adaptive allocation.

This confirms that efficiency gains translate to practical runtime improvements, not merely token-level savings.

## F.3  METRIC ROBUSTNESS: $\mathcal{E}^3$ VS. $A/T$

Our primary metric is $\mathcal{E}^3 = A^2/T$, which more strongly penalizes accuracy degradation while rewarding compute efficiency. To ensure that improvements are not metric-specific, we additionally report results under the conventional $A/T$ measure.

Table 8: Experiment results comparing $\mathcal{E}^3$ and $A/T$ across different reasoning models on MATH-500.

| | Models →
Methods↓ | DeepSeek-R1-Distill-Qwen-32B | QwQ-32B | DeepSeek-R1-Distill-Llama-70B | o4-mini |
|---|---|---|---|---|---|
| **Direct** | Vanilla | 3.83 / 4.26 | 2.04 / 2.41 | 3.58 / 3.96 | 12.20 / 13.10 |
| | Global Budget | 5.26 / 5.87 | 3.20 / 3.53 | 4.55 / 5.01 | 13.25 / 14.43 |
| **Planned** | Vanilla | 4.40 / 4.83 | 2.23 / 2.60 | 4.20 / 4.56 | 15.65 / 17.04 |
| | Global Budget | 5.36 / 5.88 | 2.91 / 3.30 | 5.16 / 5.57 | 14.39 / 15.67 |
| **PLAN-AND-BUDGET** | + Uniform | 5.64 / 6.26 | 3.28 / 3.70 | 5.41 / 5.86 | 15.88 / 17.38 |
| | + Weighted | 5.51 / 6.09 | 3.08 / 3.53 | 5.51 / 5.95 | 15.60 / 17.03 |
| | + Linear | **6.07 / 6.74** | 3.31 / 3.76 | 5.57 / 6.03 | 15.34 / 16.94 |
| | + Exponential | 5.93 / 6.53 | 3.33/ 3.79 | **5.89 / 6.33** | 15.72 / 17.29 |
| | + Polynomial | 5.91 / 6.56 | 3.32 / 3.76 | 5.58 / 6.07 | 15.55 / 17.21 |
| | + Cosine | 5.92 / 6.58 | **3.40 / 3.84** | 5.80 / 6.24 | **15.95 / 17.46** |

Table 9: Experiment results comparing $\mathcal{E}^3$ and $A/T$ across different reasoning models on NaturalInstructions.

| | Models →
Methods↓ | DeepSeek-R1-Distill-Qwen-32B | QwQ-32B | DeepSeek-R1-Distill-Llama-70B | o4-mini |
|---|---|---|---|---|---|
| **Direct** | Vanilla | 1.95 / 4.49 | 1.02 / 2.37 | 2.08 / 4.82 | 4.84 / 10.25 |
| | Global Budget | 2.33 / 5.44 | 1.47 / 3.29 | 2.48 / 5.67 | 4.88 / 10.75 |
| **Planned** | Vanilla | 2.10 / 4.93 | 1.13 / 2.61 | 2.29 / 5.28 | 5.56 / 12.69 |
| | Global Budget | 2.52 / 5.92 | 1.55 / 3.50 | 2.61 / 6.14 | 5.34 / 12.20 |
| **PLAN-AND-BUDGET** | + Uniform | 2.61 / 6.36 | **1.75 / 3.99** | 2.78 / 6.47 | **5.57 / 12.64** |
| | + Weighted | 2.57 / 6.22 | 1.65 / 3.80 | **2.96 / 6.87** | 5.14 / 11.76 |
| | + Linear | 2.73 / 6.56 | 1.72 / 3.97 | 2.88 / 6.86 | 5.37 / 12.16 |
| | + Exponential | 2.64 / 6.37 | 1.70 / 3.92 | 2.93 / 6.86 | 5.23 / 11.97 |
| | + Polynomial | **2.86 / 6.91** | 1.70 / 3.87 | 2.91 / 6.73 | 5.48 / 12.29 |
| | + Cosine | 2.73 / 6.60 | 1.69 / 3.86 | 2.79 / 6.51 | 5.42 / 12.22 |

Table 10: Experiment results comparing $\mathcal{E}^3$ and $A/T$ across different reasoning models on TravelPlanner.

| | Models →
Methods↓ | DeepSeek-R1-Distill-Qwen-32B | QwQ-32B | DeepSeek-R1-Distill-Llama-70B | o4-mini |
|---|---|---|---|---|---|
| **Direct** | Vanilla | 0.14 / 1.00 | 0.35 / 1.02 | 0.50 / 1.93 | 0.086 / 0.74 |
| | Global Budget | 0.16 / 1.19 | 0.37 / 1.22 | 0.49 / 2.00 | 0.056 / 0.67 |
| **Planned** | Vanilla | 0.30 / 1.50 | 0.38 / 1.01 | 0.64 / 2.09 | 0.091 / 0.74 |
| | Global Budget | 0.41 / 1.82 | 0.39 / 1.10 | 0.77 / 2.51 | 0.037 / 0.52 |
| **PLAN-AND-BUDGET** | + Uniform | 0.35 / 1.68 | 0.45 / 1.26 | 0.81 / 2.56 | 0.090 / 0.82 |
| | + Weighted | 0.45 / 1.91 | 0.40 / 1.19 | 0.74 / 2.48 | 0.088 / 0.81 |
| | + Linear | 0.34 / 1.72 | 0.45 / 1.29 | 0.86 / 2.72 | 0.104 / 0.89 |
| | + Exponential | 0.40 / 1.85 | 0.46 / 1.30 | 0.86 / 2.69 | 0.075 / 0.76 |
| | + Polynomial | **0.47 / 2.01** | 0.49 / 1.39 | **0.93 / 2.85** | **0.104 / 0.91** |
| | + Cosine | 0.36 / 1.77 | **0.52 / 1.43** | 0.85 / 2.70 | 0.077 / 0.78 |

Tables 8–10 show ranking stability across all methods and datasets under both metrics. The relative ranking of Plan-and-Budget over baselines remains consistent under $A/T$, confirming that the observed gains are not artifacts of metric design.

Importantly, $\mathcal{E}^3$ provides stronger discrimination in scenarios where $A/T$ appears nearly indistinguishable. For example, in several cases $A/T$ values are comparable (e.g., 6.86 vs. 6.87), while $\mathcal{E}^3$ correctly differentiates models based on accuracy preservation (e.g., 2.96 vs. 2.88). This reflects the intended design of $\mathcal{E}^3$, which penalizes accuracy drops more heavily and prevents reward hacking (e.g., achieving high $A/T$ by answering only a small subset of easy questions correctly).

Overall, the consistency across metrics reinforces the robustness of our conclusions.

Table 11: Token usage under varying budget limit for **DeepSeek-R1-Distill-Qwen-32B** on the MATH-500 dataset. The total budget for a question is given by $B_i = B_{init} + B_{per\ level} * d_i$, shrinking as $B_{init}$ and $B_{per\ level}$ decrease. $d_i$ is the question difficulty level $\in [1, 5]$. We see that the overall token usage is well correlated with the provided $B_i$.

| $B_{per\ level} \rightarrow$ 
 $B_{init} \downarrow$ | 0 | 25 | 50 | 75 | 100 |
|---|---|---|---|---|---|
| 25 | $1294.43_{\pm56.55}$ | $1323.71_{\pm64.99}$ | $1430.35_{\pm61.56}$ | $1503.34_{\pm21.12}$ | $1525.20_{\pm36.43}$ |
| 50 | $1302.50_{\pm34.40}$ | $1358.96_{\pm30.11}$ | $1442.07_{\pm28.38}$ | $1524.96_{\pm34.95}$ | $1550.97_{\pm37.36}$ |
| 75 | $1313.53_{\pm25.56}$ | $1400.21_{\pm34.00}$ | $1500.47_{\pm37.41}$ | $1528.84_{\pm35.27}$ | $1556.04_{\pm64.54}$ |
| 100 | $1359.06_{\pm46.23}$ | $1402.72_{\pm40.43}$ | $1466.34_{\pm35.83}$ | $1492.23_{\pm17.67}$ | $1572.06_{\pm44.80}$ |

Table 12: Experiment results across different reasoning models on MATH-500 (planning and budgeting by LLaMA 3.3-**70B**). Acc denotes accuracy.

| Models → | | DeepSeek-R1-Distill-Qwen-32B | | | QwQ-32B | | | DeepSeek-R1-Distill-Llama-70B | | | o4-mini | | |
|---|---|---|---|---|---|---|---|---|---|---|---|---|---|
| Methods↓ | | Acc (%) ↑ | Avg. Tok.↓ | $\mathcal{E}^3$ ↑ | Acc (%) ↑ | Avg. Tok.↓ | $\mathcal{E}^3$ ↑ | Acc (%) ↑ | Avg. Tok.↓ | $\mathcal{E}^3$ ↑ | Acc (%) ↑ | Avg. Tok.↓ | $\mathcal{E}^3$ ↑ |
| **Direct** | Vanilla | $89.76_{\pm0.26}$ | $2105.12_{\pm31.94}$ | 3.83 | $84.88_{\pm1.18}$ | $3523.72_{\pm97.42}$ | 2.04 | $90.44_{\pm0.61}$ | $2286.63_{\pm26.42}$ | 3.58 | $\mathbf{93.16_{\pm0.89}}$ | $711.20_{\pm8.31}$ | 12.20 |
| | Global Budget | $89.60_{\pm0.88}$ | $1526.15_{\pm10.09}$ | 5.26 | $\mathbf{90.56_{\pm0.33}}$ | $2565.18_{\pm37.10}$ | 3.20 | $\mathbf{90.80_{\pm0.62}}$ | $1810.83_{\pm51.64}$ | 4.55 | $91.84_{\pm0.48}$ | $636.41_{\pm8.14}$ | 13.25 |
| **Planned** | Vanilla | $\mathbf{90.12_{\pm0.39}}$ | $1633.00_{\pm43.75}$ | 4.97 | $85.56_{\pm1.37}$ | $2635.96_{\pm34.61}$ | 2.78 | $90.44_{\pm0.71}$ | $1799.48_{\pm36.42}$ | 4.55 | $91.08_{\pm1.05}$ | $534.92_{\pm17.66}$ | 15.51 |
| | Global Budget | $89.64_{\pm0.43}$ | $1377.95_{\pm22.21}$ | 5.83 | $87.48_{\pm1.08}$ | $2291.79_{\pm45.92}$ | 3.34 | $89.44_{\pm0.67}$ | $1527.37_{\pm18.04}$ | 5.24 | $91.04_{\pm0.74}$ | $578.99_{\pm4.86}$ | 14.32 |
| **PLAN-AND-BUDGET** | + Uniform | $89.44_{\pm0.52}$ | $1319.23_{\pm22.30}$ | 6.06 | $87.72_{\pm0.63}$ | $\mathbf{1982.11_{\pm64.25}}$ | 3.88 | $90.12_{\pm0.63}$ | $1513.84_{\pm43.93}$ | 5.36 | $90.56_{\pm0.38}$ | $\mathbf{518.39_{\pm17.87}}$ | 15.82 |
| | + Weighted | $89.40_{\pm0.73}$ | $1320.31_{\pm40.66}$ | 6.05 | $88.48_{\pm0.87}$ | $2041.16_{\pm38.86}$ | 3.84 | $89.92_{\pm0.30}$ | $1513.59_{\pm60.18}$ | 5.34 | $90.44_{\pm0.62}$ | $535.17_{\pm5.51}$ | 15.28 |
| | + Linear | $88.96_{\pm0.59}$ | $1294.38_{\pm48.75}$ | 6.11 | $87.60_{\pm0.72}$ | $1986.53_{\pm22.58}$ | 3.86 | $89.76_{\pm1.31}$ | $1480.78_{\pm31.96}$ | 5.44 | $89.76_{\pm0.38}$ | $525.23_{\pm12.06}$ | 15.34 |
| | + Exponential | $89.12_{\pm0.58}$ | $1348.38_{\pm37.14}$ | 5.89 | $88.12_{\pm0.66}$ | $2006.38_{\pm25.92}$ | 3.87 | $90.04_{\pm0.54}$ | $1472.01_{\pm41.92}$ | 5.51 | $90.68_{\pm0.61}$ | $522.61_{\pm11.85}$ | 15.73 |
| | + Polynomial | $89.76_{\pm0.41}$ | $\mathbf{1263.16_{\pm28.74}}$ | **6.38** | $88.24_{\pm0.61}$ | $2025.34_{\pm50.02}$ | 3.84 | $89.72_{\pm0.58}$ | $1467.00_{\pm35.92}$ | 5.49 | $89.96_{\pm0.67}$ | $525.00_{\pm10.33}$ | 15.41 |
| | + Cosine | $89.16_{\pm1.28}$ | $1304.74_{\pm48.53}$ | 6.09 | $88.84_{\pm0.75}$ | $2009.68_{\pm19.24}$ | **3.93** | $90.00_{\pm0.62}$ | $\mathbf{1462.90_{\pm43.88}}$ | **5.54** | $91.12_{\pm0.67}$ | $520.69_{\pm7.31}$ | **15.95** |

Table 13: Experiment results across different reasoning models on NaturalInstructions (planning and budgeting by LLaMA 3.3-**70B**).

| Models → | | DeepSeek-R1-Distill-Qwen-32B | | | QwQ-32B | | | DeepSeek-R1-Distill-Llama-70B | | | o4-mini | | |
|---|---|---|---|---|---|---|---|---|---|---|---|---|---|
| Methods↓ | | ROUGE (%) ↑ | Avg. Tokens ↓ | $\mathcal{E}^3$ ↑ | ROUGE (%) ↑ | Avg. Tokens ↓ | $\mathcal{E}^3$ ↑ | ROUGE (%) ↑ | Avg. Tokens ↓ | $\mathcal{E}^3$ ↑ | ROUGE (%) ↑ | Avg. Tokens ↓ | $\mathcal{E}^3$ ↑ |
| **Direct** | Vanilla | $\mathbf{43.47_{\pm0.52}}$ | $968.17_{\pm44.78}$ | 1.95 | $43.16_{\pm1.12}$ | $1818.34_{\pm24.99}$ | 1.02 | $43.13_{\pm0.76}$ | $894.46_{\pm50.69}$ | 2.08 | $\mathbf{47.24_{\pm0.31}}$ | $460.99_{\pm11.31}$ | 4.84 |
| | Global Budget | $42.81_{\pm0.39}$ | $787.25_{\pm58.17}$ | 2.33 | $\mathbf{44.77_{\pm0.73}}$ | $1360.49_{\pm101.64}$ | 1.47 | $\mathbf{43.80_{\pm1.28}}$ | $772.98_{\pm47.44}$ | 2.48 | $45.39_{\pm1.27}$ | $422.20_{\pm56.78}$ | 4.88 |
| **Planned** | Vanilla | $42.27_{\pm0.41}$ | $844.40_{\pm48.01}$ | 2.12 | $44.24_{\pm0.67}$ | $1426.74_{\pm52.92}$ | 1.37 | $43.22_{\pm0.58}$ | $799.16_{\pm18.86}$ | 2.34 | $44.06_{\pm0.54}$ | $\mathbf{346.99_{\pm8.78}}$ | 5.59 |
| | Global Budget | $42.68_{\pm0.70}$ | $711.15_{\pm23.50}$ | 2.56 | $45.13_{\pm0.56}$ | $1265.78_{\pm33.23}$ | 1.61 | $42.69_{\pm0.43}$ | $672.38_{\pm15.38}$ | 2.71 | $43.86_{\pm0.26}$ | $354.80_{\pm14.31}$ | 5.42 |
| **PLAN-AND-BUDGET** | + Uniform | $42.10_{\pm0.54}$ | $657.50_{\pm21.15}$ | 2.70 | $44.47_{\pm0.35}$ | $996.91_{\pm31.31}$ | 1.98 | $42.15_{\pm0.33}$ | $663.51_{\pm19.51}$ | 2.68 | $44.24_{\pm0.50}$ | $348.46_{\pm8.45}$ | **5.62** |
| | + Weighted | $42.32_{\pm0.65}$ | $\mathbf{620.52_{\pm45.63}}$ | **2.89** | $44.40_{\pm0.61}$ | $1025.02_{\pm24.91}$ | 1.92 | $43.14_{\pm0.84}$ | $654.49_{\pm29.22}$ | **2.84** | $44.00_{\pm0.77}$ | $363.85_{\pm7.65}$ | 5.32 |
| | + Linear | $42.22_{\pm0.49}$ | $646.42_{\pm45.63}$ | 2.76 | $44.22_{\pm0.66}$ | $1003.24_{\pm26.23}$ | 1.95 | $42.62_{\pm0.59}$ | $648.90_{\pm18.14}$ | 2.80 | $44.17_{\pm0.32}$ | $364.65_{\pm8.99}$ | 5.35 |
| | + Exponential | $42.82_{\pm1.09}$ | $645.47_{\pm20.51}$ | 2.84 | $43.99_{\pm0.22}$ | $1026.89_{\pm8.51}$ | 1.88 | $43.07_{\pm0.84}$ | $656.73_{\pm21.19}$ | 2.82 | $43.88_{\pm0.96}$ | $362.81_{\pm12.50}$ | 5.31 |
| | + Polynomial | $42.45_{\pm0.20}$ | $630.62_{\pm14.45}$ | 2.86 | $44.66_{\pm0.68}$ | $\mathbf{995.95_{\pm14.43}}$ | **2.00** | $42.45_{\pm0.38}$ | $\mathbf{641.39_{\pm16.15}}$ | 2.81 | $44.41_{\pm0.64}$ | $362.96_{\pm10.62}$ | 5.43 |
| | + Cosine | $41.96_{\pm0.46}$ | $637.09_{\pm16.98}$ | 2.76 | $44.53_{\pm0.54}$ | $1000.64_{\pm17.85}$ | 1.98 | $42.86_{\pm0.41}$ | $663.74_{\pm13.38}$ | 2.77 | $44.20_{\pm0.73}$ | $365.65_{\pm8.00}$ | 5.34 |

## F.4 BUDGET SENSITIVITY ANALYSIS

We evaluate sensitivity to the initial budget $B_{\text{init}}$ and per-level allocation $B_{\text{per-level}}$. As shown in Table 11, performance scales smoothly with increasing budget, without exhibiting abrupt degradation or instability.

These results confirm that PLAN-AND-BUDGET maintains controllable and predictable behavior across resource regimes, aligning with the budget-feasible design described in Section 4.

## F.5 PLANNER SENSITIVITY

To assess dependence on the planning component, we replace the default planner with the stronger LLaMA 3.3-**70B** and re-evaluate performance. As shown in Tables 12–14, improvements persist across planners, demonstrating that gains are not tied to a specific planning model.

This indicates that the allocation mechanism, rather than planner-specific behavior, drives the efficiency improvements.

Table 14: Experiment results on TravelPlanner (planning and budgeting by LLaMA 3.3-**70B**). Rate denotes the hard constraint pass rate.

| Models → | DeepSeek-R1-Distill-Qwen-32B | | | QwQ-32B | | | DeepSeek-R1-Distill-Llama-70B | | | o4-mini | | |
|---|---|---|---|---|---|---|---|---|---|---|---|---|
| Methods↓ | Rate (%)↑ | Avg. Tokens↓ | $\mathcal{E}^3$↑ | Rate (%)↑ | Avg. Tokens↓ | $\mathcal{E}^3$↑ | Rate (%)↑ | Avg. Tokens↓ | $\mathcal{E}^3$↑ | Rate (%)↑ | Avg. Tokens↓ | $\mathcal{E}^3$↑ |
| **Direct** Vanilla | $14.33_{\pm2.17}$ | $1430.14_{\pm43.73}$ | 0.14 | $34.89_{\pm3.20}$ | $3432.33_{\pm78.66}$ | 0.35 | $26.22_{\pm1.82}$ | $1361.37_{\pm47.93}$ | 0.50 | $11.58_{\pm2.15}$ | $1559.65_{\pm8.84}$ | 0.086 |
| Global Budget | $13.78_{\pm1.20}$ | $1158.81_{\pm20.23}$ | 0.16 | $30.78_{\pm2.06}$ | $2530.04_{\pm40.87}$ | 0.37 | $24.33_{\pm2.30}$ | $1215.29_{\pm35.05}$ | 0.49 | $8.33_{\pm1.71}$ | $\mathbf{1248.53_{\pm26.97}}$ | 0.056 |
| **Planned** Vanilla | $21.22_{\pm2.56}$ | $1379.60_{\pm58.31}$ | 0.33 | $\mathbf{34.78_{\pm3.90}}$ | $3691.57_{\pm159.20}$ | 0.33 | $33.11_{\pm3.39}$ | $1392.14_{\pm17.92}$ | 0.79 | $\mathbf{12.00_{\pm2.62}}$ | $1610.36_{\pm51.42}$ | 0.089 |
| Global Budget | $21.44_{\pm1.65}$ | $1215.41_{\pm36.19}$ | 0.38 | $34.22_{\pm2.65}$ | $3080.28_{\pm59.47}$ | 0.38 | $34.22_{\pm0.84}$ | $1248.56_{\pm49.10}$ | 0.94 | $6.89_{\pm2.50}$ | $1380.11_{\pm24.49}$ | 0.034 |
| **PLAN-AND-BUDGET** + Uniform | $23.11_{\pm2.47}$ | $1237.61_{\pm47.96}$ | 0.43 | $36.78_{\pm3.39}$ | $2668.10_{\pm110.02}$ | 0.51 | $30.44_{\pm2.02}$ | $1149.63_{\pm32.45}$ | 0.81 | $10.78_{\pm1.60}$ | $1314.86_{\pm51.88}$ | 0.088 |
| + Weighted | $\mathbf{22.89_{\pm3.48}}$ | $1208.54_{\pm68.78}$ | 0.43 | $32.56_{\pm1.15}$ | $2777.30_{\pm54.72}$ | 0.38 | $30.22_{\pm2.44}$ | $1202.07_{\pm32.48}$ | 0.76 | $10.67_{\pm3.08}$ | $1333.59_{\pm28.09}$ | 0.085 |
| + Linear | $21.89_{\pm2.53}$ | $\mathbf{1241.55_{\pm16.59}}$ | 0.39 | $35.56_{\pm2.69}$ | $2531.69_{\pm54.93}$ | 0.50 | $30.00_{\pm1.11}$ | $1162.22_{\pm45.53}$ | 0.77 | $11.11_{\pm1.71}$ | $1299.06_{\pm53.41}$ | 0.095 |
| + Exponential | $22.22_{\pm1.11}$ | $1182.30_{\pm36.35}$ | 0.42 | $32.89_{\pm2.50}$ | $2568.07_{\pm58.46}$ | 0.42 | $31.56_{\pm2.13}$ | $1156.43_{\pm30.26}$ | 0.86 | $9.56_{\pm1.54}$ | $1315.87_{\pm43.68}$ | 0.069 |
| + Polynomial | $23.44_{\pm2.82}$ | $1194.25_{\pm56.38}$ | **0.46** | $33.67_{\pm1.60}$ | $2504.92_{\pm53.37}$ | 0.45 | $\mathbf{33.00_{\pm3.03}}$ | $\mathbf{1142.54_{\pm25.70}}$ | **0.95** | $11.33_{\pm1.08}$ | $1268.11_{\pm32.39}$ | **0.101** |
| + Cosine | $22.22_{\pm2.00}$ | $1164.77_{\pm39.57}$ | 0.42 | $37.11_{\pm2.65}$ | $\mathbf{2446.04_{\pm52.80}}$ | **0.56** | $31.00_{\pm3.78}$ | $1115.86_{\pm22.65}$ | 0.86 | $9.67_{\pm1.78}$ | $1268.06_{\pm44.75}$ | 0.074 |

### F.6 TOKEN USAGE DISTRIBUTION ON TRAVELPLANNER

In addition to the answer pass rates discussed in the main results, we also examine the token usage distribution across queries of varying difficulty. Figure 4 presents the average token usage and corresponding pass rates on TravelPlanner, grouped by difficulty level.

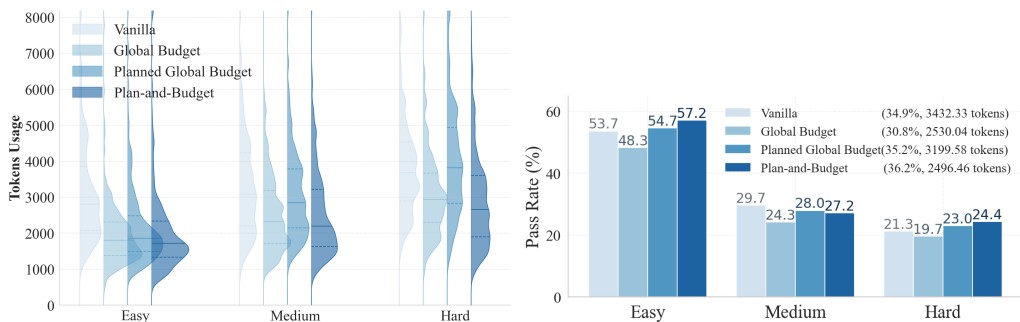

Figure 4: Token usage and pass rate analysis across difficulty levels on TravelPlanner. (Left) Token usage distributions. (Right) Answer pass rate (%) by difficulty level.

As expected, we observe that token usage increases with query difficulty across all models—more complex tasks naturally require deeper reasoning and longer responses. However, methods using global budget constraints exhibit a consistently higher token usage across all difficulty levels compared to our approach (PLAN-AND-BUDGET). This suggests that global budgeting fails to adapt to query complexity, resulting in inefficient allocation of compute.

Moreover, global methods not only consume more tokens but also suffer from reduced answer pass rates at every difficulty level. This inefficiency leads to lower overall $\mathcal{E}^3$, further reinforcing the advantage of our uncertainty-aware local budgeting strategy. In contrast, PLAN-AND-BUDGET adapts computation based on sub-question difficulty, achieving better calibration of reasoning effort across both simple and complex queries.

## G ADDITIONAL EXPERIMENTAL DETAILS

### G.1 DATASET DESCRIPTIONS AND EVALUATION METRICS

To evaluate the general applicability of our framework, we select three reasoning-heavy benchmarks spanning symbolic math, instruction following, and long-horizon planning. The dataset statistics are shown in Table 15.

**MATH-500** (Lightman et al., 2024). This is a curated 500-problem subset from the full MATH dataset, designed to test symbolic, multi-step math reasoning. Each problem requires the model

to interpret, manipulate, and solve high-school-level mathematical expressions. Performance is measured using exact-match accuracy against gold answers.

**NaturalInstructions** (Wang et al., 2022). This is a broad instruction-following benchmark consisting of over 1600 tasks covering question answering, classification, transformation, and reasoning. We randomly sample 500 test queries from the public split for evaluation. Since answers are open-ended and linguistic, we use the ROUGE score to measure semantic overlap with the reference answers.

**TravelPlanner** (Xie et al., 2024). This is a challenging planning benchmark that simulates real-world itinerary construction under hard constraints (e.g., timing or location compatibility)

Table 15: Dataset statistics. LLaMA 3.1-8B performance is also provided.

|  | MATH-500 | Natural Instructions | Travel Planner |
|---|---|---|---|
| Task | Math Reasoning | Instruction Following | Agentic Planning |
| QA Pairs | 500 | 500 | 180 |
| Metrics | Accuracy | ROUGE | Pass rate |
| LLaMA 3.1-8B Performance | $48.76_{\pm 0.74}$ | $21.72_{\pm 0.98}$ | $2.91_{\pm 0.28}$ |

and soft common sense preferences. We focus on the sole-planning setting where all relevant knowledge is embedded in the prompt, and no tool use is required. We evaluate on the validation set using the *hard constraint pass rate*, measuring whether the generated plan satisfies the minimal feasibility constraints (e.g., no overlaps or missing connections). We omit the stricter full success rate (which includes common sense and preference matching) to isolate planning competence. Prior work reports that even strong proprietary models achieve limited performance, highlighting the task's inherent complexity.

## G.2 LICENSES FOR EXISTING ASSETS

All models and datasets used in this work are publicly available and used in accordance with their respective licenses:

- **DeepSeek-R1-Distill-Qwen-32B and DeepSeek-R1-Distill-LLaMA-70B** (Guo et al., 2025) are released under the DeepSeek open-source model license available at `https://github.com/deepseek-ai/DeepSeek-LLM/blob/main/LICENSE-MODEL`.

- **QwQ-32B** (Team, 2025) is licensed under the Apache License 2.0.

- **OpenAI o4-mini** (OpenAI, 2025) is accessed via the OpenAI API under the terms of service and usage policies listed at `https://openai.com/policies/terms-of-use`. No model weights are released or modified.

- **LLaMA 3.1-8B-Instruct & LLaMA 3.3-70B-Instruct:** is used via OpenRouter and follows Meta's LLaMA 3 license available at `https://ai.meta.com/llama/license/`.

- All datasets (MATH-500, NaturalInstructions, TravelPlanner) are publicly available and properly cited. They are used for evaluation purposes under academic and research-friendly terms of use.

## G.3 COMPUTE RESOURCES

All experiments were conducted using API-accessible large language models, including OpenAI's o4-mini and models hosted via OpenRouter (e.g., DeepSeek-R1-Distill-Qwen-32B). Since our method operates entirely at inference time through prompting, the computational cost is directly proportional to the number of tokens generated. We report token usage for each setting in the main paper, which can be used to estimate wall-clock runtime given model-specific generation rates (typically 50–80 tokens/sec, depending on the provider) and the parallelism used.

All data preprocessing, prompt generation, and evaluation were performed on a cloud-based virtual machine equipped with an Intel Xeon E5-2698 CPU and 500GB of main memory. No model training or fine-tuning was involved, and the overall compute requirements are modest.

## H  PROMPT TEMPLATES

**Prompt Templates for Question Decomposition**

-Goal-
You are an experienced expert in domain and exam question designer. Your role is to help students break down challenging math problems into a series of simpler, high-level sub-questions.
We don't want too many detailed sub-questions, which are not beneficial for testing students' ability in an exam. Each sub-question should build on the previous one so that, once all have been answered, the complete solution is clear.
Your output should be a list of sub-questions with brief hints explaining the purpose of each step, but you should not reveal your internal chain-of-thought either the final solution.

Instructions for Decomposition:
First, analyze the problem and identify the key ideas needed to solve it. Then, generate a series of 2 to 5 sub-questions that lead the student step by step to the complete solution. The difficulty level of the problem is presented out of 5, where 1 is easy, and 5 is hard. Please adjust the number of sub-questions based on the level. Ideally, we want fewer sub-questions for easy problems and more sub-questions for challenging problems.
DO NOT perform reasoning, directly output those sub-questions based on your gut feelings; only output the list of sub-questions with brief hints for each.
Your answer should be a list of numbered sub-questions. Each sub-question should have a brief accompanying hint that explains what the student will achieve by answering that part.

Example Decomposition:
**Problem:** Find the remainder when $(9 \times 99 \times 999 \times \cdots \times \underbrace{99\cdots9}_{999 \text{ 9's}})$ is divided by 1000.

**Level:** 3 out of 5

**Decomposed Sub-questions:**

1. Compute the product modulo 8.
Hint: Simplify each term using $(10 \equiv 2 \mod 8)$, noting that $(10^k \equiv 0 \mod 8)$ for $k \geq 3$, leading to terms of $(-1 \mod 8)$.

2. Compute the product modulo 125.
Hint: Recognize $(10^3 \equiv 0 \mod 125)$, so terms for $(k \geq 3)$ become $(-1 \mod 125)$. Calculate the product of the first two terms and combine with the remaining terms.

3. Solve the system of congruences using the Chinese Remainder Theorem.
Hint: Combine the results from modulo 8 and modulo 125 to find a common solution modulo 1000.

A student has presented you with the following math problem:
Problem: <problem>
Level: <level> out of 5
**REMEMBER**, you are not allowed to think about it, please directly generate the answer in the following:
Decomposed Sub-questions:

**Prompt Templates for Question Difficulty Evaluation**

You are an experienced expert in <domain> and exam question designer. Your task is to evaluate the difficulty level of a given exam problem and its sub-questions by comparing it

against a set of benchmark questions of known levels.
Based on their levels, you will need to assign each subquestion a portion of the credits (assuming the total credit points is 100 for the whole problem).

Each level reflects increasing complexity from 1 (easiest) to 5 (most challenging). Evaluate based on the conceptual depth, steps involved in solving, required knowledge, and potential for misdirection.

Use the following benchmark examples as references:

<benchmarks>

1. You will be provided a question and its subquestions. You will evaluate the difficulty level of the problem and its sub-questions.
Assuming the whole problem is worth 100 points, you assign each sub-question a portion of the score points.
- Adhere to the given subquestions, and DO NOT make new subquestions.
- Sum of each subquestion's credits MUST EQUAL to 100.

2. You must return the result in a structured JSON format:
{
"problem": {"reason": "...", "evaluated_level": level_q}
"1": {"reason": "...", "evaluated_level": level_1, "credit": credit_1},
"2": {"reason": "...", "evaluated_level": level_2, "credit": credit_2},
...}
where
- "reason": a short explanation (up to 50 words) of your level assessment.
- "evaluated_level": an integer from 1 to 5 indicating your judgment.
- "credit": an integer between 1 to 100 indicating when the question is solved correctly, how many credit can be given.

Evaluate the level of the following question:
Problem: <problem>
Sub-questions: <steps>
Output:

**Prompt Templates for Vanilla Model**

< dataset-specific instruction>

Please reason step by step, and conclude your answer in the following format:

<dataset specific output format>

Question: <query>
Reference: <reference> (only applicable to TravelPlanner)
Output: <think>

**Prompt Templates for Global Budget Model**

< dataset-specific instruction>

Please reason step by step, and conclude your answer in the following format:

<dataset specific output format>

Question: <query>
Reference: <reference> (only applicable to TravelPlanner)
Let's think step by step and use less than <budget> tokens. Output: <think>

**Prompt Templates for Planned Vanilla Model**

<dataset-specific instruction>

The problem is given by an overall description, difficulty level out of 5, followed by a series of sub-questions as a hint.
All the credit is given when you provide a correct final answer for the overall problem.
Please solve the question efficiently and clearly to achieve as much credit as possible.

Let's start the exam. You are being given this math problem:
**Problem:** <query>
**Reference:** <reference>(only applicable to TravelPlanner)
**Level:** <level> out of 5

You may think following these sub-questions or feel free to use other methods that works the best towards getting the final answer:
<decomposed>

Please provide your final answer in the following format:
<dataset specific output format>

Output: <think>

**Prompt Templates for Planned Global Budget Model**

<dataset-specific instruction>

The problem is given by an overall description, difficulty level out of 5, followed by a series of sub-questions as a hint.
All the credit is given when you provide a correct final answer for the overall problem.
Please solve the question efficiently and clearly to achieve as much credit as possible.

Let's start the exam. You are being given this math problem:
**Problem:** <query>
**Reference:** <reference> (only applicable to TravelPlanner)
**Level:** <level> out of 5

You may think following these sub-questions or feel free to use other methods that works the best towards getting the final answer:
<decomposed>

Please provide your final answer in the following format:
<dataset specific output format>

Let's think step by step and use less than <budget> tokens.
Output: <think>

**Prompt Templates for Planned Local Budget Model (Ours)**

<dataset-specific instruction>

The problem is given by an overall description, difficulty level out of 5, followed by a series of sub-questions as a hint.
All the credit is given when you provide a correct final answer for the overall problem.
Please solve the question efficiently and clearly to achieve as much credit as possible.

Let's start the exam. You are being given this math problem:
**Problem:** <query>
**Reference:** <reference> (only applicable to TravelPlanner)
**Level:** <level> out of 5

You may think following these sub-questions or feel free to use other methods that works the best towards getting the final answer:
<decomposed> (For each decomposed subquestion:) Please only think a little, and directly solve it using up to <budget> words.

Please provide your final answer strictly following the format:
<dataset specific output format>

Output: <think>

