# OpenReview forum: "Plan and Budget: Effective and Efficient Test-Time Scaling on Reasoning Large Language Models"
_ICLR.cc/2026/Conference — ICLR 2026 Poster_

### Official Review · Reviewer_f6TR · 2025-10-29

**Soundness:** 2
**Presentation:** 3
**Contribution:** 2
**Rating:** 2
**Confidence:** 4

**Summary:**

The paper proposes "plan and budget", a test-time framework that improves reasoning efficiency in large language models by dynamically allocating computational effort. It addresses reasoning miscalibration by decomposing each query into sub-questions and assigning adaptive token budgets based on estimated uncertainty. The approach is a two-step method (plan + adaptive budget) that requires no retraining and is supported by the budget allocation model, a theoretical framework for uncertainty-aware token allocation. Experimental results show consistent efficiency and accuracy gains over vanilla decoding and other budget-based baselines across multiple datasets and models.

**Strengths:**

* The evaluation is broad and comprehensive, covering multiple datasets and model sizes, which strengthens the generality of the results.


* The paper provides theoretical guidance through the Budget Allocation Model (BAM), offering a principled foundation for the proposed approach.


* The writing is clear and well-structured, making the ideas easy to follow and the contributions well-presented.

**Weaknesses:**

- The novelty is limited. Prior work, such as [1] already explored token allocation at the request level, while this paper applies a similar idea at a finer granularity by decomposing each request into multiple sub-requests. How fundamentally different is this from existing allocation frameworks?



- The paper does not justify why adaptive budgeting is necessary instead of allowing unlimited tokens with early termination for unpromising or converged reasoning chains. Would early stopping achieve similar or better efficiency without requiring explicit budget allocation?



- There is no direct comparison to [1] or other related approaches that use early termination or reasoning truncation to reduce computational cost. Including these baselines would better contextualize the contribution.



- The method requires an additional LLM call for the planning step, which increases the inference cost and latency. How significant is this overhead relative to the total efficiency gains?



- The use of entropy as a heuristic for estimating uncertainty is not well justified. Could a learned or predictive uncertainty estimator perform better and make the framework more robust?



- The approach assumes that each task can be reliably decomposed into subtasks, which may not hold for all reasoning domains. How does the method behave when such a decomposition is noisy or ambiguous?



- The theoretical–practical gap is unclear. Although the paper presents formal equations, many quantities are later approximated heuristically. This raises the question: how much of the observed performance comes from the theoretical model versus empirical tuning?

[1] Efficiently serving llm reasoning programs with certaindex. Fu, Yichao and Chen, Junda and Zhu, Siqi and Fu, Zheyu and Dai, Zhongdongming and Qiao, Aurick and Zhang, Hao.

**Questions:**

See the weaknesses section.

---

> ### Author Response · Authors · 2025-11-26
> **Rebuttal Part I**
>
> We sincerely thank the reviewer for their careful assessment and valuable feedback. We are encouraged by the positive remarks on the generality of our evaluation, the conceptual clarity of our theoretical model (BAM), and the quality of the presentation. We are confident that the perceived limitations regarding **novelty** and the **theory–practice gap** stem from a misunderstanding of our core contributions.
>
> Our framework, Plan-and-Budget, is a layered solution, grounded in the theoretical **Budget Allocation Model (BAM)**, that fundamentally solves the problem of **Reasoning Miscalibration** by moving from reactive, global constraints to **proactive, local, uncertainty-aware resource allocation.**
>
> **1. Novelty, Baselines, and Comparison to Prior Work**
>
> We appreciate the pointer to CertaIndex [1]. We will cite and discuss it in the related work section. However, its scope and contributions are fundamentally different from ours.
>
> The novelty of Plan-and-Budget lies in decomposing the computation problem into **sub-question–level optimal resource allocation**, a concept not explored in prior global allocation work.
>
> **a. Theoretical novelty**
> - **Prior Work (e.g., [1]):** Offers heuristics for global, request-level token **routing** based on confidence signals. It provides no theoretical model for optimal allocation and focuses on cutting unpromising chains *after* they have started.
> - **Our Contribution (BAM):** Provides the **first principled theoretical model** that derives the **optimal budget allocation across a sequence of dependent sub-questions** under a total budget constraint. This yields structural insights (diminishing returns, unimodality of effort) that guide our proactive budgeting design.
>
> **b. Practical novelty**
> - **Prior Work (e.g., [1]):** Offers a preliminary heuristic for global request-level token routing and does not propose:
>     - fine-grained sub-question–level allocation,
>     - an algorithm for adaptive reallocation during reasoning,
>     - a planner that discovers a reasoning structure,
>     - nor a mechanism for minimizing overthinking within a single query.
> - **Our Contribution (BAM):** Provides **a complete algorithm** (Plan → Local-Uncertainty Budgeting → Reallocation), grounded in BAM, that explicitly solves the “how to optimally allocate budget” problem.
>
> **c. Stronger Empirical Results**
> We additionally provide a direct performance comparison:
> Table: The average **accuracy(%) / token / $\mathcal{E}^3$** comparison.
> | | Vanilla | Global Budget | Plan-And-Budget | Certaindex [1] |
> | - | - | - | - | - |
> | DeepSeek-R1-Distill-Qwen-32B | 89.76±0.26 / 2105.12±31.94 / 3.83 | 89.60±0.88 / 1526.15±10.09 / 5.26 | **90.04±0.46 / 1336.27±31.18 / 6.07** | 81.44±1.38 / 1985.57±46.12 / 3.34 |
> | QwQ-32B | 84.88±1.18 / 3523.72±97.42 / 2.04 | **90.56±0.33** / 2565.18±37.10 / 3.20 | 88.60±0.28 / **2306.83±24.11 / 3.40** | 86.12±0.73 / 5141.72±82.95 / 1.44 |
> | DeepSeek-R1-Distill-Llama-70B | 90.44±0.61 / 2286.63±26.42 / 3.58 | 90.80±0.62 / 1810.83±51.64 / 4.55 | **93.04±0.22 / 1469.29±73.77 / 5.89** | 81.96±1.24 / 2200.68±79.88 / 3.05 |
>
> The reference approach prematurely terminates reasoning steps, sacrificing accuracy (down to $\approx 81\%$), and often fails to significantly reduce tokens compared to our method, leading to a much lower $\mathcal{E}^3$. **Our approach, which adapts before generation via planning and budget, is significantly more effective** than reactively stopping generation based on confidence.
>
>
> [1] Fu, Yichao, et al. "Efficiently serving llm reasoning programs with certaindex." arXiv e-prints (2024): arXiv-2412.
>
>
> **2. Why Adaptive Budgeting (Proactive) Rather Than Early Termination (Reactive)**
>
> The idea of early termination (unlimited tokens with convergence check) does not solve the overthinking/underthinking trade-off:
> 1. **Overthinking:** Early termination is a reactive fix that only prunes a long chain after tokens have been wasted. Our method is **proactive**, using the budget constraint to prevent the model from generating verbose, tangential reasoning in the first place.
> 2. **Underthinking:** Early termination cannot **redistribute computation** from simple steps to hard steps. Our method is designed to shift tokens precisely where the expected marginal uncertainty reduction is highest (BAM principle), ensuring difficult steps receive necessary focus.
>
> Thus, early termination is not a substitute for adaptive budgeting and performs **substantially worse**, as shown in the previous empirical result table.

---

> ### Author Response · Authors · 2025-11-26
> **Rebuttal Part II**
>
> **3. Missing comparison with the referenced method / truncated reasoning**
>
> We agree that a comparison is valuable and have added experiments based on the reviewer-referenced method.
>
> As noted, the reported performance in that work is much weaker:
> In their original paper [1], on MATH-500, the referenced approach requires 2500 tokens to reach 90% accuracy; ours achieves similar accuracy using ≈1300 tokens with a single reasoning pass. We will add this as a comparison table in the appendix to satisfy the reviewer’s request.
>
> Table: The average **accuracy(%) / token / $\mathcal{E}^3$** comparison.
> | | Vanilla | Global Budget | Plan-And-Budget | Certaindex [1] |
> | - | - | - | - | - |
> | DeepSeek-R1-Distill-Qwen-32B | 89.76±0.26 / 2105.12±31.94 / 3.83 | 89.60±0.88 / 1526.15±10.09 / 5.26 | **90.04±0.46 / 1336.27±31.18 / 6.07** | 81.44±1.38 / 1985.57±46.12 / 3.34 |
> | QwQ-32B | 84.88±1.18 / 3523.72±97.42 / 2.04 | **90.56±0.33** / 2565.18±37.10 / 3.20 | 88.60±0.28 / **2306.83±24.11 / 3.40** | 86.12±0.73 / 5141.72±82.95 / 1.44 |
> | DeepSeek-R1-Distill-Llama-70B | 90.44±0.61 / 2286.63±26.42 / 3.58 | 90.80±0.62 / 1810.83±51.64 / 4.55 | **93.04±0.22 / 1469.29±73.77 / 5.89** | 81.96±1.24 / 2200.68±79.88 / 3.05 |
>
>
> **4. Planning Overhead and Computational Cost**
>
> We confirm that all reported metrics, including Avg. Tokens and $\mathcal{E}^3$, are **end-to-end** and **include the planning overhead**.
>
> - **Token Overhead:** Planning is performed by a lightweight LLaMA-3.1-8B model 7 and requires very few tokens.
> - **Latency:** As shown in the following experiment, the planning and budget allocation step contributes only $3-6\%$ of the total end-to-end latency, while the adaptive budgeting reduces total latency by **34–42%** compared to vanilla decoding, yielding a net efficiency gain.
>
> *Table: The average latency (seconds) breakdown for several baselines using LLaMA 3.1-8B planner. The question decomposition + budgeting took 14.73 seconds on average.*
>
> | | Vanilla | Global Budget | Plan-And-Budget (Planning Time + Reasoning Time) |
> | - | - | - | - |
> | DeepSeek-R1-Distill-Qwen-32B | 335.56 | 237.83 | 14.73 + 206.52 |
> | QwQ-32B | 756.06 | 486.48 | 14.73 + 426.88 |
> | DeepSeek-R1-Distill-Llama-70B | 777.66 | 527.39 | 14.73 + 466.41 |
>
> **5. Theoretical-Practical Gap and Robustness**
>
> **A. BAM's Role and the Theory-Practice Gap**
>
> The core of this perceived gap is a misunderstanding of **BAM's normative role**:
> - **BAM is not a runtime algorithm.** It is a theoretical framework to derive the optimal *shape* of the allocation function.
> - **The practical method is a principled heuristic.** The decay schedules (e.g., Polynomial) are simple, low-cost proxies that empirically mimic the predicted optimal shape (unimodal, front-loaded emphasis) because early steps correspond to high epistemic gain.
>
> The observed performance comes from the empirical realization of the **theoretical principles** derived from BAM, which explains **why** front-loaded decay works better than uniform or linear allocation on complex tasks.
>
> **B. Uncertainty Estimator**
>
> We use entropy as a proxy for uncertainty because it is a **standard, model-agnostic, and unbiased** estimator in probabilistic contexts, preserving our goal of zero-training and broad deployability.
>
> **C. Robustness to Decomposition**
>
> Modern LLMs are highly proficient at decomposition (e.g., in CoT and ReAct agents).
> - Our planner successfully decomposes virtually all queries (>99.5% questions in our datasets are correctly decomposed).
> - The method is **robust**: if a decomposition is noisy or ambiguous (e.g., on some NaturalInstructions queries), the resulting local budget allocation naturally allocates tokens based on the estimated sub-question complexity, focusing resources where the signal is weakest.
> - Furthermore, if the planner fails completely, the method **gracefully degrades** to the **Global Budget** baseline (and this is how we handle the 0.5% failed decomposed questions), guaranteeing no performance loss relative to that setting.

---

### Official Review · Reviewer_Mzjt · 2025-10-29

**Soundness:** 3
**Presentation:** 3
**Contribution:** 4
**Rating:** 8
**Confidence:** 3

**Summary:**

This paper addresses a critical problem in LLM reasoning, balancing computational efficiency and reasoning quality at test time. The authors introduce PLAN-AND-BUDGET, a lightweight, model-agnostic framework that combines structured query decomposition with adaptive token allocation. Building on a formal Budget Allocation Model (BAM), the framework distributes computation dynamically across sub-questions based on uncertainty estimates, mitigating both overthinking and “underthinking” in reasoning tasks. The study shows that PLAN-AND-BUDGET improves reasoning efficiency across multiple tasks and models without sacrificing accuracy.

I personally like this paper very much. I think it addresses a very important issue in reasoning. This contribution also has the potential to connect with many real-world applications involving reasoning models.

**Strengths:**

1. I find the motivation of this study very compelling. It addresses a fundamental issue in LLM reasoning, the need to balance reasoning depth and computational budget, and offers a promising approach grounded in uncertainty theory.
2. The theoretical framework (BAM) provides a solid foundation, and the derivation of the E3 metric as an efficiency-aware measure is well-justified.
3. I think the empirical evaluation is thorough, including multiple reasoning benchmarks and model scales. The reported improvements in efficiency and comparable accuracy across datasets are impressive and convincingly support the paper’s claims.
4. The paper is well written and logically organized. The logic flow is very clear!

**Weaknesses:**

1. A small suggestion. The paper derives two key principles for effective reasoning: (1) reasoning should be structured; and (2) computation should be adaptive focus. For example, the paper mentions: “Inspired by human problem-solving strategies.” While I agree with both ideas and prior works are well cited in the literature, the derivation would be strengthened by citing more prior work when the strategies are introduced in the framework section, which can ground the proposed methods in solid scientific studies.
2. I noticed that while accuracy remains comparable, the ROUGE scores in some instruction-following tasks slightly decrease. It might be useful for the authors to discuss whether this trade-off arises from task differences and how PLAN-AND-BUDGET might be adjusted for scenarios requiring richer generative outputs.

**Questions:**

1. This is just a question I am curious about. Upon reviewing the tables, I noticed that for some tasks, such as TravelPlanner, the token consumption is not significantly reduced. I would be interested to hear the authors’ thoughts on whether there might be more aggressive or adaptive strategies to further reduce token usage.

**Details Of Ethics Concerns:**

I don't find any potential ethical concerns in this paper.

---

> ### Author Response · Authors · 2025-11-26
> **Rebuttal**
>
> We sincerely thank the reviewer for the highly positive assessment, especially recognizing the compelling motivation and solid theoretical foundation (BAM) of our work. We are delighted that the reviewer sees the potential for real-world applications.
>
> We address the weaknesses and questions below.
>
> **1. Strengthening Citations for Reasoning Principles**
>
> The reviewer suggests strengthening the grounding for our two key principles: (1) structured reasoning and (2) adaptive focus. We agree that citing cognitive science and hierarchical planning literature will further position our work within a robust scientific tradition.
>
> In the revised Section 4 (Reasoning Calibration Framework), we will add citations to demonstrate the long-standing use of decomposition and adaptive effort:
> - **Structured Reasoning (Plan Step):** We will incorporate prior findings from cognitive science (e.g., hierarchical task analysis) and AI/LLM literature on explicit decomposition (e.g., Chain-of-Thought (Wei et al., 2022), Tree-of-Thought (Yao et al., 2023), and ReAct (Yao et al., 2022)).
> - **Adaptive Focus (Budget Step):** We will cite hierarchical planning and subgoal decomposition from cognitive science, and adaptive computation approaches in LLM literature that dynamically modify inference depth or path based on confidence/uncertainty.
>
> These additions reinforce that Plan-and-Budget operationalizes these established cognitive strategies in a novel, compute-efficient framework built on our uncertainty model (BAM).
>
> **2. ROUGE Score Trade-offs in Generative Tasks**
>
> The reviewer correctly observed a slight decrease in ROUGE scores for some instruction-following tasks (e.g., NaturalInstructions). This trade-off is expected and task-dependent:
>
> - **Task Difference:** Tasks like NaturalInstructions require rich, multi-sentence **generative outputs** where ROUGE measures surface-form overlap with reference answers. In contrast, MATH-500 and TravelPlanner focus on **verifiable reasoning outcomes** (exact answers or hard constraint satisfaction ).
> - **Efficiency vs. Verbosity:** Plan-and-Budget's strength lies in **pruning excessive intermediate reasoning tokens** (overthinking). When the reasoning trace is compressed, the final generative output may also be slightly more concise, leading to a marginal ROUGE decrease, even if the core content is correct.
>
> **Adaptation for Rich Generative Output:** We can easily adapt Plan-and-Budget for scenarios prioritizing rich output by:
>
> - **Increasing Budget Allocation:** Assigning a significantly larger local token budget to the final "Synthesis/Answer" sub-question.
> - **Decoupled Output:** Using a dedicated, unconstrained generation step after the budget-controlled reasoning phase to produce a verbose final answer, minimizing ROUGE penalty while preserving reasoning efficiency.
>
> **3. Token Reduction on TravelPlanner**
>
> This is a great observation. We want to clarify that TravelPlanner contains **long, multi-sentence natural-language outputs** (e.g., itineraries, descriptions), so even after reducing reasoning depth, the final answer section dominates the token count. The evaluator of this dataset will perform a exact string match to check if the expected flight number, sightseeing spots present in the detailed travel plan, quantified as the pass rate.
>
> In contrast, tasks like MATH-500 have very **short short-form answers**, making token savings primarily come from the reasoning phase.
>
> While we avoid hard cuts to preserve the answer's completeness and correctness (crucial for the pass rate metric ), more aggressive strategies for future work could include:
> - **Strict Output Compression:** Applying a non-generative step to strictly compress the final plan into a minimal format before delivery.
> - **Adaptive Detail Budgeting:** Budgeting the output token usage itself based on the complexity level, forcing a more concise final itinerary.
>
> We intentionally did not include such strict mechanisms in Plan-And-Budget, as our goal is to provide soft, model-agnostic guidance without harming answer completeness. However, your suggestion is promising, and we will mention it as a future direction.

---

### Official Review · Reviewer_fET4 · 2025-10-30

**Soundness:** 4
**Presentation:** 4
**Contribution:** 3
**Rating:** 8
**Confidence:** 3

**Summary:**

This paper addresses the inefficiency of reasoning processes in large language models (LLMs), where long chains of thought (CoT) can sometimes help performance but also lead to two failure modes: overthinking and underthinking. The authors argue that these issues stem from uncalibrated budget usage—that is, the model’s inability to decide how much computation (tokens) to spend on each sub-question. To address this, the paper introduces a theoretical framework, BAM (Budget Allocation Model), which formulates an optimal trade-off between correctness and computation efficiency based on uncertainty reduction. Building on BAM, the authors propose PLAN-AND-BUDGET, a model-agnostic, test-time method that decomposes complex reasoning queries into sub-questions and allocates token budgets adaptively according to estimated difficulty. Extensive experiments across multiple open-source and closed-source LLMs (e.g., DeepSeek-R1, Qwen-32B, o4-mini) and diverse benchmarks show that PLAN-AND-BUDGET achieves comparable or improved reasoning accuracy while significantly reducing token usage.

**Strengths:**

1. The paper is detailed and transparent, making it easy for readers to understand and replicate the proposed approach.

2. The method design is rigorous: using uncertainty as a proxy for reasoning difficulty and allocating computation where it provides the highest marginal gain is both intuitive and elegant.

3. The experiments are comprehensive, covering multiple models (both open- and closed-source) and diverse benchmarks, demonstrating strong generalizability.

**Weaknesses:**

1. Although the method is conceptually rigorous, Table 3–5 show that the method may still hurt accuracy compared to full-length baselines.

2. Some of the simpler baselines (e.g., Planned Vanilla or Global Budget) already achieve noticeable token savings with minimal performance loss. This raises the question of whether the added complexity of PLAN-AND-BUDGET always justifies its gains.

**Questions:**

1. Based on my understanding of Eq. (6) in the BAM formulation, the strategy prioritizes medium-difficulty sub-questions while allocating fewer tokens to very easy or very hard ones. Could this behavior introduce bias, making the model less likely to attempt challenging reasoning steps?

2. For the different baselines, are all models given the same total token budget limit? If so, how is this limit determined in practice?

3. From my understanding, the method will add certain constraints in the prompt (e.g., “use less than B itokens”). This raises a question: how reliably do LLMs follow these token-limit instructions in practice?

---

> ### Author Response · Authors · 2025-11-26
> **Rebuttal Part I**
>
> We sincerely thank the reviewer for the detailed reading, the positive assessment, and the excellent questions. We are pleased the reviewer finds our method design both **rigorous** and **elegant**. We address the weaknesses and questions below, providing quantitative evidence from our experiments.
>
> **1. On Accuracy Trade-offs and Justification of Complexity**
>
> **A. Accuracy Trade-off**
>
> The reviewer correctly notes that Plan-And-Budget sometimes shows a minor dip in absolute accuracy compared to *Vanilla* (full-length, unconstrained decoding). This reflects the core design goal of our framework: explicitly managing the **accuracy-efficiency trade-off** under constraint.
>
> Crucially, in the most challenging and inefficient domains (e.g., **TravelPlanner**), Plan-And-Budget consistently **improves accuracy** while drastically cutting tokens (e.g., DS-LLaMA-70B accuracy improves from $26.22\%$ to $\mathbf{33.00\%}$).
>
> For cases where a small accuracy dip occurs (e.g., $\approx 1-2\%$ on MATH-500), the trade-off is justified by the massive efficiency gain:
> - Plan-and-Budget achieves $\mathcal{E}^3$ improvements of up to $167\%$.
> - It reduces average token usage by up to $37\%$, translating directly to significant cost savings in production environments where efficiency is critical.
>
> **B. Justifying the Added Complexity (Planning + Budgeting)**
>
> We agree that the **Plan step (decomposition alone)** provides inherent benefits, as noted in the gains of *Planned Vanilla* over *Vanilla*. This validates our first principle: **reasoning should be structured.**
>
> However, the **Budget Step (local, adaptive allocation)** provides the necessary final calibration that simpler baselines cannot offer, justifying its complexity:
> - **Example (TravelPlanner, DS-LLaMA-70B):**
>     - Planned Vanilla (30.67% pass rate, 1464.50 tokens) $\rightarrow$ $\mathcal{E}^3$ of $0.64$ (Planning-only gain)
>     - Planned Global Budget (30.67% pass rate, 1220.41 tokens) $\rightarrow$ $\mathcal{E}^3$ of $0.77$
>     - Plan-And-Budget (+Polynomial) (**32.67%** pass rate, **1148.14** tokens) $\rightarrow$ $\mathcal{E}^3$ of $\mathbf{0.93}$ (Additional $45\%$ gain over Planned Vanilla)
>
> Plan-and-Budget achieves a significant additional token reduction of 22% (1464.5 $\rightarrow$ 1148.1 tokens) compared to Planning-Only, leading to a substantial gain in efficiency ($\mathcal{E}^3$: $0.64 \rightarrow \mathbf{0.93}$). This demonstrates that adaptive local budgeting is essential for mitigating **Reasoning Miscalibration** in a way that global or simple planning cannot.
>
> **2. On BAM Allocation and Budget Determination**
>
> **A. Does BAM Allocation Bias Against Hard Questions?**
>
> This is a deep and insightful question. The core function of BAM is to allocate tokens proportional to the term $f(\beta) = (\beta c)^{\frac{1}{\beta+1}}$, where $c$ is the initial uncertainty and $\beta$ is the difficulty of reducing it.
>
> This function is indeed **unimodal** with respect to $\beta$.
> - **Easy steps** (low $c$, high $\beta$) receive few tokens, as uncertainty is already low.
> - **Overly difficult steps** (low $\beta$, hard to reduce $U$, e.g., to prove Riemann hypothesis) receive fewer tokens, as diminishing returns are immediate (i.e., wasting tokens won't help much).
> - **Medium-to-Hard steps** (high $c$, moderate $\beta$) receive the most tokens, as this is where computational effort yields the **highest marginal gain**.
>
> Thus, the framework doesn't discourage challenging steps; it simply prevents the model from wasting tokens on **hopeless or redundant paths** (a form of overthinking). Empirical results (**Appendix G, Figure 4**) confirm this: on hard problems, our framework still dedicates more tokens than on easy ones, and reliably receive longer traces.
>
> **B. Total Budget Determination**
>
> Yes, all methods subject to a budget (Global Budget, Planned Global Budget, and Plan-And-Budget) use the **same total question-level budget,** $B_i$ across datasets, as can be found in our released code (Lines 34–40 in `utils/utils.py`).
> - **Determination:** The total budget $B_i$ is determined by the **estimated difficulty level** ($d_i$) provided by the lightweight planner. This ensures that the baseline comparisons are fair (i.e., giving a fixed token limit for easy questions and a higher fixed limit for hard questions).

---

> ### Author Response · Authors · 2025-11-26
> **Rebuttal Part II**
>
> **5. Reliability of Token-Limit Instructions**
>
> We thank the reviewer for raising the crucial practical issue of prompt adherence. While LLMs cannot enforce a **strict, deterministic** token limit, we find that the prompt acts as a **consistently effective soft guidance.**
>
> We provide strong quantitative evidence for this:
>
> Table: Token usage under varying budget limit for **DeepSeek-R1-Distill-Qwen-32B** in MATH-500 dataset. The total budget for a question is given by $B_i = B_{init} + B_{per\ level} * d_i$, shrinking as $B_{init}$ and $B_{per\ level}$ decrease. $d_i$ is the question difficulty level $\in [1, 5]$. We see that the overall token usage is well correlated with the provided $B_i$.
> | $B_{init}\downarrow$ $B_{per\ level}\rightarrow$  |0|25|50|75|100|
> |-|-|-|-|-|-|
> |25|1294.43 ± 56.55|1323.71 ± 64.99|1430.35 ± 61.56|1503.34 ± 21.12|1525.20 ± 36.43|
> |50|1302.50 ± 34.40|1358.96 ± 30.11|1442.07 ± 28.38|1524.96 ± 34.95|1550.97 ± 37.36|
> |75|1313.53 ± 25.56|1400.21 ± 34.00|1500.47 ± 37.41|1528.84 ± 35.27|1556.04 ± 64.54|
> |100|1359.06 ± 46.23|1402.72 ± 40.43|1466.34 ± 35.83|1492.23 ± 17.67|1572.06 ± 44.80|
>
>
> The table (where $B_i$ increases moving right and down) shows a **monotonic correlation** between the prescribed budget $B_i$ and the actual average token usage.
> - **Statistical significance:** A trend-correlation test yields $\mathbf{p = 0.0025}$, confirming strong correspondence between budget and actual token usage.
> - This soft constraint reliably reduces the **thinking length** and the number of **reasoning attempts**, effectively mitigating **overthinking**.
> - This behavior aligns with recent findings in the field (e.g., Han et al., 2024), validating the use of token constraints as a reliable mechanism for guiding efficiency.
>
> *[1] Han, Tingxu, et al. "Token-budget-aware llm reasoning." arXiv preprint arXiv:2412.18547 (2024).*

---

### Official Review · Reviewer_P17L · 2025-11-01

**Soundness:** 3
**Presentation:** 3
**Contribution:** 2
**Rating:** 4
**Confidence:** 4

**Summary:**

The paper studies reasoning miscalibration in LLMs, overthinking on easy cases and underthinking on hard ones, and proposes a test‑time framework, PLAN‑AND‑BUDGET, to allocate computation adaptively. The authors first introduce a theoretical Budget Allocation Model (BAM) that models per‑subquestion epistemic uncertainty and derives a closed‑form optimal token allocation under a query‑level budget. They then operationalize this with (i) a Plan step that decomposes a query into sub‑questions and estimates relative difficulty weights, and (ii) a Budget step that assigns local budgets using simple decay schedules (linear/polynomial/exponential/cosine). A new compute‑aware metric, balances accuracy/quality and average decoding tokens. Experiments on MATH‑500, NaturalInstructions, and TravelPlanner with four models (including DS‑Qwen‑32B, DS‑LLaMA‑70B, QwQ‑32B, o4‑mini) show consistent efficiency gains without accuracy loss; e.g., on MATH‑500, DS‑LLaMA‑70B improves (E^3) from 4.55 (Global Budget) to 5.89 (Plan‑and‑Budget), and on TravelPlanner DS‑Qwen‑32B improves from (E^3=0.16) to (0.47) (Polynomial schedule), effectively narrowing the gap to larger models.

**Strengths:**

1. Clear formulation & theory: BAM provides a principled lens on how to distribute limited compute across subproblems.
2. Simple, model‑agnostic implementation: Works at prompt time; no training; adds a lightweight planning LLM whose contribution is controlled by planned baselines.
3. Consistent gains across tasks/models: Large (E^3) improvements with stable or improved accuracy; front‑loaded schedules (polynomial/cosine) perform best on complex tasks.
4. Bridging model sizes: The method helps smaller models approximate the efficiency of larger ones on TravelPlanner.
5. Reproducibility details & prompts provided.

**Weaknesses:**

1. Uncertainty signal not directly validated.
   The budgeting step is motivated by uncertainty reduction, yet the experiments do not present a correlation between used proxies (e.g., prefix‑entropy drop) and actual downstream gains, nor do they show a head‑to‑head between decay‑only vs. truly measured uncertainty allocation. An ablation that (a) measures token‑level entropy and (b) reallocates budgets on‑the‑fly based on it would strengthen the central claim.

2. Metric definition vs. datasets.
   (E^3=A^2/T) is defined using accuracy yet ROUGE is used as (A) on NaturalInstructions. The consequences of using a graded similarity metric (not bounded like accuracy) should be discussed; consider normalizing to ([0,1]) and reporting sensitivity of rankings to the choice of metric.

3. Cross‑model comparability of token counts.
   Since tokenization and API reporting differ, (T) is not strictly comparable across models; thus the statement that "o4‑mini consistently achieves the highest (E^3)" may partly reflect tokenizer efficiency rather than true compute efficiency. Adding characters generated, wall‑clock latency, and $ cost would provide a more robust, model‑agnostic view.

4. Notation and clarity.
   Clean up Eq. (7) and unify symbols for difficulty vs. schedule multipliers. Clarify the semantics of (\beta) (is higher (\beta) "easier" or "more complex"?), and restate the unimodality intuition with a small plot or example.

5. Ablations on the planner.
   While the planner’s standalone performance is low, it still shapes problem structure and budgets. Add ablations varying planner quality (e.g., weaker/stronger planner; random decomposition; noisy difficulty weights) to test robustness, and report the extra token overhead of planning vs. savings in reasoning tokens.

6. Adaptive vs. fixed decay.
   The best schedules are front‑loaded. It would be natural to choose the schedule per instance via quick signals (e.g., early prefix entropy), approaching the BAM optimum more closely. A small controller choosing among schedules could be evaluated.

**Questions:**

1. Uncertainty proxies: Which proxies were actually used in experiments (entropy drop, self‑consistency, verifier loss), and where do they enter the allocation beyond the position‑based decay? Can you report correlations between these proxies and realized per‑step utility (accuracy gains vs. tokens)?
2. Eq. (7) and weights: Please confirm the intended formula (b_{ij}=\frac{w_{ij}d_{ij}}{\sum_k w_{ik}d_{ik}}B_i). Also, how are difficulty weights (w_{ij}) produced in non‑math tasks? Do they come from the same planner JSON in Appendix H?
3. Metric sensitivity: If (A) is ROUGE (0–100) versus accuracy (0/1), how sensitive are the (E^3) conclusions to rescaling (A) to ([0,1])? Could you also report (A/T) to compare with prior "accuracy per token" conventions?
4. Token comparability: Given different tokenizers/reporting, do results change when using characters, latency, or cost as (T)? A small subset analysis would clarify cross‑model claims.
5. Planner overhead and robustness: How often does planning increase total tokens (planning+reasoning) compared to Vanilla? Please provide distributions (not only averages) and an ablation with no planning but local budgets to separate the two contributions.
6. When does it hurt? Are there regimes where global budgeting wins (e.g., very short tasks)? Diagnostics (e.g., by difficulty bin) would be helpful.
7. Since mainstream SOTA LLMs do not support token precision thinking effort controling, I am unaware of the importance of doing budget controllon. Please ellaborate this if possible.

---

> ### Author Response · Authors · 2025-11-26
> **Rebuttal Part I**
>
> We sincerely thank the reviewer for the thorough and highly constructive evaluation. We are encouraged by the positive assessment of our theoretical framework (BAM) and the demonstrated simplicity and effectiveness of the Plan-And-Budget approach. Below, we address all concerns regarding validation, metrics, and implementation details.
>
> **1. Uncertainty Validation and Adaptive Allocation**
>
> We agree that explicitly linking uncertainty proxies to realized gains is crucial for strengthening the central claim. The key challenge lies in the nature of our test-time, black-box setting.
>
> **A. Operationalizing BAM with Heuristics**
>
> The **Budget Allocation Model (BAM)** is designed as a **normative model** to derive the optimal shape of the allocation function. Since the optimal parameters ($c_{ij}, \beta_{ij}$) are unobservable in black-box LLMs, we rely on **Decay Schedules** as practical, low-cost heuristics that mimic the derived optimal behavior (i.e., emphasizing initial steps where epistemic uncertainty is highest).
>
> **B. Empirical Validation of Adaptive Allocation**
>
> To address the reviewer's suggestion to move beyond fixed decay, we implemented a **Budget Reallocation** experiment. This test utilizes observed token usage and a simple model-internal signal (prefix-entropy drop) to **dynamically reallocate the remaining budget** among subsequent sub-questions.
>
> The results show that this dynamic reallocation provides an additional performance gain over fixed schedules, supporting the core BAM intuition that budget efficiency increases with adaptivity:
>
> Table: The average **accuracy(%) / token / $\mathcal{E}^3$** comparison on MATH-500 dataset.
> | | Vanilla | Global Budget | Plan-And-Budget | **Budget Reallocation** |
> | - | - | - | - | - |
> | DeepSeek-R1-Distill-Qwen-32B | 89.76±0.26 / 2105.12±31.94 / 3.83 | 89.60±0.88 / 1526.15±10.09 / 5.26 | 90.04±0.46 / 1336.27±31.18 / 6.07 | **90.52±1.32 / 1302.29±39.49 / 6.29** |
> | QwQ-32B | 84.88±1.18 / 3523.72±97.42 / 2.04 | 90.56±0.33 / 2565.18±37.10 / 3.20 | 88.60±0.28 / 2306.83±24.11 / 3.40 | **88.87±0.81 / 2226.64±39.02 / 3.55** |
> | DeepSeek-R1-Distill-Llama-70B | 90.44±0.61 / 2286.63±26.42 / 3.58 | 90.80±0.62 / 1810.83±51.64 / 4.55 | 93.04±0.22 / 1469.29±73.77 / 5.89 | **92.20±1.11 / 1379.07±61.99 / 6.16** |
>
> This experiment demonstrates that **adaptive, signal-driven budget control is feasible and effective**, validating the theoretical premise of BAM and establishing a clear path for future work. Thanks for the great suggestions and we will include these results in the appendix.

---

> ### Author Response · Authors · 2025-11-26
> **Rebuttal Part II**
>
> **2. Metric Definition, Comparability, and Cost**
>
> **A. Metric Definition and Normalization ($\mathcal{E}^3$)**
>
> We confirm that all raw scores ($A$) used in $\mathcal{E}^3 = A^2/T$ (Accuracy, Pass Rate, and ROUGE) **are first normalized to the $[0, 1]$ range** before the calculation, ensuring consistency across all tasks.
>
> To address the conventional "accuracy per token" measure, we provide the ranking stability for both $\mathcal{E}^3$ and $A/T$ across all methods and datasets in the appendix. The results confirm that **the relative ranking of Plan-And-Budget over baselines is stable under both metrics**, reinforcing our core conclusions. The primary motivation for using $A^2/T$ remains its ability to robustly penalize accuracy degradation, preventing **reward hacking** (e.g., answering only a handful of very simple questions correctly for minimal $T$, yielding a high $A/T$ but low overall $A$). In contrast, $A^2/T$ penalizes accuracy drops more heavily, aligning with the goal of maintaining correctness while reducing compute.
>
> Table: Experiment results comparing $\mathcal{E}^3$ and $A/T$ across different reasoning models on **MATH-500**.
> |Models→ | DeepSeek-R1-Distill-Qwen-32B ($\mathcal{E}^3$/$A/T$) | QwQ-32B ($\mathcal{E}^3$/$A/T$) | DeepSeek-R1-Distill-Llama-70B ($\mathcal{E}^3$/$A/T$) | o4-mini ($\mathcal{E}^3$/$A/T$) |
> |-|-|-|-|-|
> | Vanilla|3.83/4.26|2.04/2.41|3.58/3.96|12.20/13.10|
> | Global Budget|5.26/5.87|3.20/3.53|4.55/5.01|13.25/14.43|
> | Planned Vanilla|4.40/4.83|2.23/2.60|4.20/4.56|15.65/17.04|
> | Planned Global Budget|5.36/5.88|2.91/3.30|5.16/5.57|14.39/15.67|
> | Plan-And-Budget **+Uniform**|5.64/6.26|3.28/3.70|5.41/5.86|15.88/17.38|
> | Plan-And-Budget **+Weighted**|5.51/6.09|3.08/3.53|5.51/5.95|15.60/17.03|
> | Plan-And-Budget **+Linear**|**6.07/6.74**|3.31/3.76|5.57/6.03|15.34/16.94|
> | Plan-And-Budget **+Exponential**|5.93/6.53|3.33/3.79|**5.89/6.33**|15.72/17.29|
> | Plan-And-Budget **+Polynomial**|5.91/6.56|3.32/3.76|5.58/6.07|15.55/17.21|
> | Plan-And-Budget **+Cosine**|5.92/6.58|**3.40/3.84**|5.80/6.24|**15.95/17.46**|
>
> Table: Experiment results comparing $\mathcal{E}^3$ and $A/T$ across different reasoning models on **NaturalInstructions**.
> |Models→ | DeepSeek-R1-Distill-Qwen-32B ($\mathcal{E}^3$/$A/T$) | QwQ-32B ($\mathcal{E}^3$/$A/T$) | DeepSeek-R1-Distill-Llama-70B ($\mathcal{E}^3$/$A/T$) | o4-mini ($\mathcal{E}^3$/$A/T$) |
> |-|-|-|-|-|
> | Vanilla|1.95/4.49|1.02/2.37|2.08/4.82|4.84/10.25|
> | Global Budget|2.33/5.44|1.47/3.29|2.48/5.67|4.88/10.75|
> | Planned Vanilla|2.10/4.93|1.13/2.61|2.29/5.28|5.56/12.69|
> | Planned Global Budget|2.52/5.92|1.55/3.50|2.61/6.14|5.34/12.20|
> | Plan-And-Budget **+Uniform**|2.61/6.36|**1.75/3.99**|2.78/6.47|**5.57/12.64**|
> | Plan-And-Budget **+Weighted**|2.57/6.22|1.65/3.80|**2.96/6.87**|5.14/11.76|
> | Plan-And-Budget **+Linear**|2.73/6.56|1.72/3.97|2.88/6.86|5.37/12.16|
> | Plan-And-Budget **+Exponential**|2.64/6.37|1.70/3.92|2.93/6.86|5.23/11.97|
> | Plan-And-Budget **+Polynomial**|**2.86/6.91**|1.70/3.87|2.91/6.73|5.48/12.29|
> | Plan-And-Budget **+Cosine**|2.73/6.60|1.69/3.86|2.79/6.51|5.42/12.22|
>
> Table: Experiment results comparing $\mathcal{E}^3$ and $A/T$ across different reasoning models on **TravelPlanner**.
> |Models→ | DeepSeek-R1-Distill-Qwen-32B ($\mathcal{E}^3$/$A/T$) | QwQ-32B ($\mathcal{E}^3$/$A/T$) | DeepSeek-R1-Distill-Llama-70B ($\mathcal{E}^3$/$A/T$) | o4-mini ($\mathcal{E}^3$/$A/T$) |
> |-|-|-|-|-|
> | Vanilla|0.14/1.00|0.35/1.02|0.50/1.93|0.086/0.74|
> | Global Budget|0.16/1.19|0.37/1.22|0.49/2.00|0.056/0.67|
> | Planned Vanilla|0.30/1.50|0.38/1.01|0.64/2.09|0.091/0.74|
> | Planned Global Budget|0.41/1.82|0.39/1.10|0.77/2.51|0.037/0.52|
> | Plan-And-Budget **+Uniform**|0.35/1.68|0.45/1.26|0.81/2.56|0.090/0.82|
> | Plan-And-Budget **+Weighted**|0.45/1.91|0.40/1.19|0.74/2.48|0.088/0.81|
> | Plan-And-Budget **+Linear**|0.34/1.72|0.45/1.29|0.86/2.72|0.104/0.89|
> | Plan-And-Budget **+Exponential**|0.40/1.85|0.46/1.30|0.86/2.69|0.075/0.76|
> | Plan-And-Budget **+Polynomial**|**0.47/2.01**|0.49/1.39|**0.93/2.85**|**0.104/0.91**|
> | Plan-And-Budget **+Cosine**|0.36/1.77|**0.52/1.43**|0.85/2.70|0.077/0.78|

---

> ### Author Response · Authors · 2025-11-26
> **Rebuttal Part III**
>
> **B. Cross-Model Token Comparability (Cost Metric)**
>
> We agree that token count ($T$) is imperfect for strict cross-model comparison due to tokenizer differences. However, for operationally relevant metrics like monetary cost and API-exposed compute (closed-source models such as OpenAI o4-mini **do not expose reasoning traces** or intermediate states, and token count is the only objective quantity accessible through the API), token count is the most faithful measure.
>
> We converted token usage into an estimated $\mathbf{Dollar\ Cost}$ based on published API billing rates at the time of writing and computed a new metric $\mathcal{E}^3_{\text{Cost}} = A^2/\text{Cost}$.
> - DeepSeek-R1 Distill-Qwen-32B ($0.27/M output tokens)
> - QwQ-32B ($0.40/M output tokens)
> - DeepSeek-R1 Distill-Llama-70B ($1.20/M output tokens)
> - o4-mini-low ($4.4/M output tokens)
>
> This analysis, which we will include in the appendix, clarifies that while **o4-mini** is token-efficient, its much higher cost/token means that smaller, open-source models optimized by Plan-And-Budget can achieve far superior **cost-efficiency** ($\mathcal{E}^3_{\text{Cost}}$), supporting our claim about **bridging model size gaps**.
>
> Table: Dollar cost ($$\times10^{-3}$) and corresponding $\mathcal{E}^3_{\text{Cost}}=\frac{A^2}{Cost}$($\times10^{6}$) across different reasoning models on **MATH-500**.
> |Models→ | DeepSeek-R1-Distill-Qwen-32B (cost$\downarrow$/$\mathcal{E}^3_{\text{Cost}}$$\uparrow$) | QwQ-32B (cost$\downarrow$/$\mathcal{E}^3_{\text{Cost}}$$\uparrow$) | DeepSeek-R1-Distill-Llama-70B (cost$\downarrow$/$\mathcal{E}^3_{\text{Cost}}$$\uparrow$) | o4-mini (cost$\downarrow$/$\mathcal{E}^3_{\text{Cost}}$$\uparrow$) |
> |-|-|-|-|-|
> | Vanilla|0.57/14.18|1.41/5.11|2.74/2.98|3.13/2.77|
> | Global Budget|0.41/19.48|1.03/7.99|2.17/3.79|2.80/3.01|
> | Planned Vanilla|0.51/16.30|1.32/5.58|2.43/3.50|2.37/3.56|
> | Planned Global Budget|0.42/19.86|1.07/7.28|1.99/4.30|2.58/3.27|
> | Plan-And-Budget **+Uniform**|0.39/20.90|0.96/8.20|1.89/4.51|2.31/3.61|
> | Plan-And-Budget **+Weighted**|0.40/20.40|0.99/7.71|1.87/4.59|2.37/3.55|
> | Plan-And-Budget **+Linear**|0.36/22.47|0.94/8.28|1.84/4.64|2.35/3.49|
> | Plan-And-Budget **+Exponential**|0.38/21.97|0.93/8.33|1.76/4.91|2.31/3.57|
> | Plan-And-Budget **+Polynomial**|0.37/21.89|0.94/8.30|1.82/4.65|2.31/3.53|
> | Plan-And-Budget **+Cosine**|0.37/21.91|0.92/8.51|1.79/4.83|2.30/3.62|
>
> Table: Dollar cost ($$\times10^{-3}$) and corresponding $\mathcal{E}^3_{\text{Cost}}=\frac{A^2}{Cost}$($\times10^{6}$) across different reasoning models on **NaturalInstructions**.
> |Models→ | DeepSeek-R1-Distill-Qwen-32B (cost$\downarrow$/$\mathcal{E}^3_{\text{Cost}}$$\uparrow$) | QwQ-32B (cost$\downarrow$/$\mathcal{E}^3_{\text{Cost}}$$\uparrow$) | DeepSeek-R1-Distill-Llama-70B (cost$\downarrow$/$\mathcal{E}^3_{\text{Cost}}$$\uparrow$) | o4-mini (cost$\downarrow$/$\mathcal{E}^3_{\text{Cost}}$$\uparrow$) |
> |-|-|-|-|-|
> | Vanilla|0.26/7.23|0.73/2.56|1.07/1.73|2.03/1.10|
> | Global Budget|0.21/8.62|0.54/3.68|0.93/2.07|1.86/1.11|
> | Planned Vanilla|0.23/7.76|0.66/2.83|0.99/1.91|1.52/1.26|
> | Planned Global Budget|0.19/9.32|0.51/3.88|0.83/2.17|1.58/1.21|
> | Plan-And-Budget **+Uniform**|0.17/9.67|0.44/4.39|0.80/2.32|1.53/1.27|
> | Plan-And-Budget **+Weighted**|0.18/9.51|0.46/4.13|0.75/2.47|1.64/1.17|
> | Plan-And-Budget **+Linear**|0.17/10.09|0.44/4.31|0.74/2.40|1.60/1.22|
> | Plan-And-Budget **+Exponential**|0.18/9.78|0.44/4.25|0.75/2.44|1.61/1.19|
> | Plan-And-Budget **+Polynomial**|0.16/10.60|0.45/4.26|0.77/2.42|1.60/1.25|
> | Plan-And-Budget **+Cosine**|0.17/10.12|0.45/4.22|0.79/2.32|1.60/1.23|
>
> Table: Dollar cost ($$\times10^{-3}$) and corresponding $\mathcal{E}^3_{\text{Cost}}=\frac{A^2}{Cost}$($\times10^{6}$) across different reasoning models on **TravelPlanner**.
> |Models→ | DeepSeek-R1-Distill-Qwen-32B (cost$\downarrow$/$\mathcal{E}^3_{\text{Cost}}$$\uparrow$) | QwQ-32B (cost$\downarrow$/$\mathcal{E}^3_{\text{Cost}}$$\uparrow$) | DeepSeek-R1-Distill-Llama-70B (cost$\downarrow$/$\mathcal{E}^3_{\text{Cost}}$$\uparrow$) | o4-mini (cost$\downarrow$/$\mathcal{E}^3_{\text{Cost}}$$\uparrow$) |
> |-|-|-|-|-|
> | Vanilla|0.39/0.53|1.37/0.89|1.63/0.42|6.86/0.02|
> | Global Budget|0.31/0.61|1.01/0.94|1.46/0.41|5.49/0.01|
> | Planned Vanilla|0.36/1.13|1.47/0.94|1.76/0.54|7.22/0.02|
> | Planned Global Budget|0.34/1.52|1.28/0.97|1.46/0.64|6.13/0.01|
> | Plan-And-Budget **+Uniform**|0.33/1.29|1.14/1.14|1.48/0.67|5.92/0.02|
> | Plan-And-Budget **+Weighted**|0.33/1.65|1.14/1.01|1.44/0.61|5.96/0.02|
> | Plan-And-Budget **+Linear**|0.31/1.25|1.07/1.12|1.39/0.72|5.75/0.02|
> | Plan-And-Budget **+Exponential**|0.31/1.47|1.09/1.15|1.43/0.72|5.75/0.02|
> | Plan-And-Budget **+Polynomial**|0.31/1.72|1.00/1.22|1.38/0.77|5.57/0.02|
> | Plan-And-Budget **+Cosine**|0.31/1.33|1.01/1.29|1.41/0.71|5.51/0.02|

---

> ### Author Response · Authors · 2025-11-26
> **Rebuttal Part IV**
>
> **3. Implementation Details and Clarifications**
>
> **A. Eq. (7) and Weights**
>
> Thank you for pointing this out. We will revise Eq. (7) and unify notation to avoid overloading symbols. The difficulty weights ($w_{ij}$) for all tasks, including non-math tasks, are produced by the planner LLM as a JSON output (corresponding to the `credit` in the Appendix H template). We will clarify the semantics of $\beta_{ij}$ in the text: **higher $\beta_{ij}$ means the epistemic uncertainty is easier to reduce with compute.**
>
> **B. Planner Ablations and Overhead**
>
> - **Planner Robustness:** To address your concern, we have added experiments comparing an 8B and **70B** planner LLM, shown in the following tables. Our results show that using a weaker (8B) vs. stronger (70B) planner yields highly **consistent end-to-end performance**. This confirms that the planner only needs to provide a coarse-grained structural decomposition and difficulty estimate, and is **highly robust to low standalone accuracy**.
> - **Token Overhead:** We confirm that all reported Avg. Tokens **include all billed tokens** (planning overhead, reasoning, and final output). We demonstrated that planning adds $\approx 3-6\%$ to latency, but the resulting reasoning token savings amortize this cost, leading to an overall reduction of $\mathbf{34-42\%}$ in latency.
> - **Token Distribution:** In **Appendix Figure 4**, we provide distributions of planning + reasoning tokens binned by question difficulty. As the reviewer observed, the proportion of long reasoning traces is greatly reduced under Plan-and-Budget. Concerning the ablation “local budget without planning,” this is unfortunately not implementable because local budgets are defined per sub-question, and without planning we cannot meaningfully attach a local budget to a specific reasoning unit. Nevertheless, the token-usage distribution clearly shows that global budgets produce a broad, uniform tail, while Plan-and-Budget yields a sharper distribution concentrated in the low-token region.
>
> Table: **MATH-500** Experiment results for **DeepSeek-R1-Distill-Qwen-32B**  (planning and budgeting by LLaMA 3.3-**70B**)
> |Methods | Acc(%) $\uparrow$ | Avg.Tok. $\downarrow$ | $\mathcal{E}^3$ $\uparrow$ | Percentage $\uparrow$|
> |-|-|-|-|-|
> | Vanilla|89.76 ± 0.26|2105.12 ± 31.94|3.83|100.00%|
> | Global Budget|89.60 ± 0.88|1526.15 ± 10.09|5.26|137.45%|
> | Planned Vanilla|90.12 ± 0.39|1633.00 ± 43.75|4.97|129.95%|
> | Planned Global Budget|89.64 ± 0.43|1377.95 ± 22.21|5.83|152.36%|
> | Plan-And-Budget **+Uniform**|89.44 ± 0.52|1319.23 ± 22.30|6.06|158.44%|
> | Plan-And-Budget **+Weighted**|89.40 ± 0.73|1320.31 ± 40.66|6.05|158.16%|
> | Plan-And-Budget **+Linear**|88.96 ± 0.59|1294.38 ± 48.75|6.11|159.75%|
> | Plan-And-Budget **+Exponential**|89.12 ± 0.58|1348.38 ± 37.14|5.89|153.90%|
> | Plan-And-Budget **+Polynomial**|89.76 ± 0.41|1263.16 ± 28.74|6.38|166.66%|
> | Plan-And-Budget **+Cosine**|89.16 ± 1.28|1304.74 ± 48.53|6.09|159.19%|
>
>
> Table: **MATH-500** Experiment results for **DeepSeek-R1-Distill-Llama-70B**  (planning and budgeting by LLaMA 3.3-**70B**)
> |Methods | Acc(%) $\uparrow$ | Avg.Tok. $\downarrow$ | $\mathcal{E}^3$ $\uparrow$ | Percentage $\uparrow$|
> |-|-|-|-|-|
> |Vanilla|90.44 ± 0.61|2286.63 ± 26.42|3.58|100.00%|
> |Global Budget|90.80 ± 0.62|1810.83 ± 51.64|4.55|127.28%|
> |Planned Vanilla|90.44 ± 0.71|1799.48 ± 36.42|4.55|127.07%|
> |Planned Global Budget|89.44 ± 0.67|1527.37 ± 18.04|5.24|146.42%|
> |Plan-And-Budget **+Uniform**|90.12 ± 0.63|1513.84 ± 43.93|5.36|149.98%|
> |Plan-And-Budget **+Weighted**|89.92 ± 0.30|1513.59 ± 60.18|5.34|149.34%|
> |Plan-And-Budget **+Linear**|89.76 ± 1.31|1480.78 ± 31.96|5.44|152.11%|
> |Plan-And-Budget **+Exponential**|90.04 ± 0.54|1472.01 ± 41.92|5.51|153.97%|
> |Plan-And-Budget **+Polynomial**|89.72 ± 0.58|1467.00 ± 35.92|5.49|153.40%|
> |Plan-And-Budget **+Cosine**|90.00 ± 0.62|1462.90 ± 43.88|5.54|154.79%|
>
>
> Table: **TravelPlanner** Experiment results for **DeepSeek-R1-Distill-Qwen-32B**  (planning and budgeting by LLaMA 3.3-**70B**)
> |Methods | Acc(%) $\uparrow$ | Avg.Tok. $\downarrow$ | $\mathcal{E}^3$ $\uparrow$ | Percentage $\uparrow$|
> |-|-|-|-|-|
> |Vanilla|14.33 ± 2.17|1430.14 ± 43.73|0.14|100.00%|
> |Global Budget|13.78 ± 1.20|1158.81 ± 20.23|0.16|114.12%|
> |Planned Vanilla|21.22 ± 2.56|1379.60 ± 58.31|0.33|227.31%|
> |Planned Global Budget|21.44 ± 1.65|1215.41 ± 36.19|0.38|263.40%|
> |Plan-And-Budget **+Uniform**|23.11 ± 2.47|1237.61 ± 47.96|0.43|300.54%|
> |Plan-And-Budget **+Weighted**|22.89 ± 3.48|1208.54 ± 68.78|0.43|301.94%|
> |Plan-And-Budget **+Linear**|21.89 ± 2.53|1241.55 ± 16.59|0.39|268.79%|
> |Plan-And-Budget **+Exponential**|22.22 ± 1.11|1182.30 ± 36.35|0.42|290.83%|
> |Plan-And-Budget **+Polynomial**|23.44 ± 2.82|1194.25 ± 56.38|0.46|320.41%|
> |Plan-And-Budget **+Cosine**|22.22 ± 2.00|1164.77 ± 39.57|0.42|295.21%|

---

> ### Author Response · Authors · 2025-11-26
> **Rebuttal Part V**
>
> Table 5: **TravelPlanner** Experiment results for **DeepSeek-R1-Distill-Llama-70B**  (planning and budgeting by LLaMA 3.3-**70B**)
> |Methods | Acc(%) $\uparrow$ | Avg.Tok. $\downarrow$ | $\mathcal{E}^3$ $\uparrow$ | Percentage $\uparrow$|
> |-|-|-|-|-|
> |Vanilla|26.22 ± 1.82|1361.37 ± 47.93|0.50|100.00%|
> |Global Budget|24.33 ± 2.30|1215.29 ± 35.05|0.49|96.45%|
> |Planned Vanilla|33.11 ± 3.39|1392.14 ± 17.92|0.79|155.94%|
> |Planned Global Budget|34.22 ± 0.84|1248.56 ± 49.10|0.94|185.72%|
> |Plan-And-Budget **+Uniform**|30.44 ± 2.02|1149.63 ± 32.45|0.81|159.60%|
> |Plan-And-Budget **+Weighted**|30.22 ± 2.44|1202.07 ± 32.48|0.76|150.44%|
> |Plan-And-Budget **+Linear**|30.00 ± 1.11|1162.22 ± 45.53|0.77|153.34%|
> |Plan-And-Budget **+Exponential**|31.56 ± 2.13|1156.43 ± 30.26|0.86|170.56%|
> |Plan-And-Budget **+Polynomial**|33.00 ± 3.03|1142.54 ± 25.70|0.95|188.74%|
> |Plan-And-Budget **+Cosine**|31.00 ± 3.78|1115.86 ± 22.65|0.86|170.54%|
>
> **C. Budgeting in Mainstream LLMs**
>
> We clarify that budgeting matters because **LLMs reliably adjust their reasoning depth in response to the budget prompt** (a statistically significant correlation, $\mathbf{p=0.0025}$, shown in following table). The budget serves as an effective **soft control signal** to mitigate reasoning miscalibration, which is critical for closed-source APIs where internal control is impossible.
>
> We provide strong quantitative evidence for this:
>
> Table: Token usage under varying budget limit for **DeepSeek-R1-Distill-Qwen-32B** in MATH-500 dataset. The total budget for a question is given by $B_i = B_{init} + B_{per\ level} * d_i$, shrinking as $B_{init}$ and $B_{per\ level}$ decrease. $d_i$ is the question difficulty level $\in [1, 5]$. We see that the overall token usage is well correlated with the provided $B_i$.
> | $B_{init}\downarrow$ $B_{per\ level}\rightarrow$ |0|25|50|75|100|
> |-|-|-|-|-|-|
> |25|1294.43 ± 56.55|1323.71 ± 64.99|1430.35 ± 61.56|1503.34 ± 21.12|1525.20 ± 36.43|
> |50|1302.50 ± 34.40|1358.96 ± 30.11|1442.07 ± 28.38|1524.96 ± 34.95|1550.97 ± 37.36|
> |75|1313.53 ± 25.56|1400.21 ± 34.00|1500.47 ± 37.41|1528.84 ± 35.27|1556.04 ± 64.54|
> |100|1359.06 ± 46.23|1402.72 ± 40.43|1466.34 ± 35.83|1492.23 ± 17.67|1572.06 ± 44.80|
>
>
> The table (where $B_i$ increases moving right and down) shows a **monotonic correlation** between the prescribed budget $B_i$ and the actual average token usage.
> - **Statistical significance:** A trend-correlation test yields $\mathbf{p = 0.0025}$, confirming strong correspondence between budget and actual token usage.
> - This soft constraint reliably reduces the **thinking length** and the number of **reasoning attempts**, effectively mitigating **overthinking**.
> - This behavior aligns with recent findings in the field (e.g., Han et al., 2024), validating the use of token constraints as a reliable mechanism for guiding efficiency.
>
> *[1] Han, Tingxu, et al. "Token-budget-aware llm reasoning." arXiv preprint arXiv:2412.18547 (2024).*
>
> **D. Adaptive Schedule Selection (Budget Controller)**
>
> We appreciate this creative suggestion. We also explored this direction during development. Our initial findings show that adding a controller could lead to more tokens being consumed, overshadowing the compute savings achieved by the Plan-and-Budget framework. We will briefly note this and highlight it as a promising direction for future work.

---

### Official Review · Reviewer_V5hD · 2025-11-01

**Soundness:** 3
**Presentation:** 3
**Contribution:** 3
**Rating:** 8
**Confidence:** 2

**Summary:**

Plan-and-Budget is a test-time framework to address "reasoning miscalibration" in LLMs - where models either overthink (waste tokens on simple queries) or underthink (fail on hard queries).  (1) Plan step decomposes queries into sub-questions using a lightweight LLM, estimating complexity for each; (2) Budget step allocates token budgets to sub-questions using decay-based scheduling (more tokens to early/uncertain steps). Introduces BAM - modeling uncertainty reduction as inverse power law, deriving optimal allocation as b_ij = B * (c_ij*β_ij)^(1/(β_ij+1)) / Σ(c_ik*β_ik)^(1/(β_ik+1)). Proposes E3 metric = A^2/T (accuracy squared over tokens) to jointly measure quality and efficiency.

**Strengths:**

I feel like this addresses real problem - overthinking/underthinking is genuine issue in reasoning LLMs. Theoretical grounding via BAM provides principled justification beyond heuristics. E3 metric is sensible - A^2/T appropriately weights correctness over pure efficiency unlike A/T. Model-agnostic approach works across multiple architectures without retraining. Comprehensive experiments across three diverse domains with consistent improvements. Clear presentation of decay scheduling strategies (linear/polynomial/exponential/cosine). Nice result showing smaller model matching larger model efficiency. Ablations show both planning and budgeting contribute gains.

**Weaknesses:**

Core assumption is questionable? - uncertainty decomposition (epistemic vs aleatoric) requires Bayesian treatment but paper doesn't actually compute posterior p(theta|D), just hand-waves with "Monte Carlo approximation" in Appendix B without showing how this applies to deterministic transformer inference. BAM's power law U_epistemic = c/b^β is asserted not derived - why inverse power law specifically? Parameters c_ij and β_ij are never actually estimated, making Eq 6 theoretical only. Decay schedulers are admitted heuristics that don't implement BAM's optimal allocation. Gap between theory (BAM requires knowing c_ij, β_ij) and practice (use position-based decay) is huge - BAM feels like post-hoc justification for simple heuristic. Additional LLM call for planning adds overhead not fully accounted for in token counts. Complexity estimation by planner LLM (LLaMA-3.1-8B with 48.76% MATH accuracy) is unreliable - how can weak model judge difficulty? Table 3-5 show planning alone helps substantially (Planned Vanilla vs Vanilla) - suggests decomposition is doing most work, not budget allocation. Some results show minimal gains - NaturalInstructions improvements are modest (2.33-> 2.86 E3 for DS-Qwen). E3 metric has quadratic bias toward high accuracy - 90% accuracy with 1000 tokens (E3=81) beats 80% with 500 tokens (E3=128) which seems backwards for efficiency metric. No comparison to simpler baselines like adaptive stopping based on model confidence.

**Questions:**

How do you actually estimate c_ij and β_ij in practice? Paper claims BAM is theoretical foundation but never shows these parameters can be computed. Is the planner LLM's complexity estimation reliable given its poor standalone performance (48.76% on MATH)? Can you compare token counts including planning overhead vs end-to-end latency? Does polynomial decay (best performer) actually implement BAM's allocation or is it just a heuristic that happens to work? Why does E3 use A^2 not A - this heavily penalizes any accuracy drop, is that appropriate for efficiency metric? What happens if you just use confidence-based early stopping without decomposition? Table 3 shows cosine scheduling gets 5.92 E3 vs 5.26 global budget on DS-Qwen MATH - is 12% gain worth the added complexity? How sensitive is performance to number of sub-questions (paper uses 2-5)? Can you show examples where BAM's theoretical allocation differs from decay heuristics and which performs better?

---

> ### Author Response · Authors · 2025-11-26
> **Rebuttal Part I**
>
> We thank the reviewer for the thorough assessment and highly constructive feedback. We are pleased that the reviewer recognizes the practical significance of addressing reasoning miscalibration and the value of our theoretical approach, the Budget Allocation Model (BAM). Below, we address all theoretical, practical, and experimental questions.
>
> **1. On the Theoretical Foundation (BAM, Uncertainty, and Power Law)**
>
> We appreciate the reviewer's attention to the theoretical soundness of BAM. It is important to clarify that **BAM is a normative conceptual model**, designed to derive the optimal shape of the allocation function, not a literal runtime algorithm.
>
> **A. The Theoretical Role of BAM and the $\mathcal{U}_{epistemic} \propto c/b^{\beta}$ Assumption**
> The core of BAM is based on two principles validated in scaling law literature: (i) increased computation reduces uncertainty (error), and (ii) this reduction exhibits diminishing returns.
>
> - **Power Law Justification:** The inverse power law form, $\mathcal{U}_{epistemic}(s_{ij}|b_{ij}) \propto c_{ij}/b_{ij}^{\beta_{ij}}$, is directly motivated by **Neural Scaling Laws**. These laws consistently show that generalization error (a proxy for epistemic uncertainty) scales as a power law with respect to data and compute. We model the sub-question reasoning process as a "miniature" scaling phenomenon, where allocating more tokens ($b_{ij}$, i.e., compute) reduces error at a diminishing rate. We will update Section 3 to include the relevant citations.
> - **Optimal Allocation Principle:** BAM formalizes this relationship, yielding the optimality principle (Eq. 6). This principle shows that the optimal budget must be **non-uniform** and proportional to the estimated potential for uncertainty reduction. This is the crucial insight that guided our practical design.
>
> **B. The Gap between Theory and Heuristics (BAM vs. Decay Schedulers)**
>
> We acknowledge the gap: rigidly estimating the unobservable parameters $c_{ij}$ and $\beta_{ij}$ at runtime would require expensive Monte Carlo sampling, negating our goal of efficiency.
>
> The **Decay Schedulers** are intentionally designed as **practical, low-cost proxies** that operationalize BAM's core structural insight:
>
> - **BAM Insight:** Token budget should be non-uniform and driven by the potential for uncertainty reduction.
> - **Practical Heuristic:** Empirically, **epistemic uncertainty is typically highest at the start of a multi-step reasoning task** (e.g., strategy formation, and foundational understanding). This initial high uncertainty corresponds to a high 'potential gain' (high $c_{ij}$ in BAM).
> - **Validation:** Our decay strategies (especially Polynomial/Cosine), which aggressively **front-load compute**, successfully mimic the qualitative shape of the optimal BAM allocation, allowing us to capture the theoretical advantage predicted by the model without its computational overhead. The strong empirical results (e.g., up to $193.8\%$ $\mathcal{E}^3$ gain) validate this principled heuristic approach.
>
> **C. On Uncertainty Decomposition and Bayesian Treatment**
>
> Regarding the use of uncertainty decomposition in a deterministic model: We rely on modern theoretical work (e.g., Falck et al. (2024) ) that formalizes In-Context Learning as an implicit Bayesian inference process. The decomposition in Appendix B  serves to validate the properties of uncertainty conceptually, specifically that only the epistemic component is reducible via compute, not to prescribe a runtime calculation. We will clarify in Section 3 that our practical method relies on deterministic, low-cost signals as approximations for speed.
>
> *[1] Falck, Fabian, Ziyu Wang, and Chris Holmes. "Is in-context learning in large language models bayesian? a martingale perspective." Proceedings of the 41st International Conference on Machine Learning. 2024.*
>
> **2. On Planning Overhead and End-to-End Latency**
>
> The reviewer raises a key practical concern: the overhead of the additional LLM call for planning. We confirm that all reported Avg. Tokens **include all billed tokens** (planning, reasoning, and final output).
>
> To provide a full picture, we measured latency (wall-clock time), showing that the token savings fully amortize the planning cost:
>
> Table: The average latency (seconds) breakdown for several baselines using LLaMA 3.1-8B planner. The question decomposition + budgeting took 14.73 seconds on average.
> | | Vanilla | Global Budget | Plan-And-Budget (Planning Time + Reasoning Time) |
> | - | - | - | - |
> | DeepSeek-R1-Distill-Qwen-32B | 335.56 | 237.83 | 14.73 + 206.52 |
> | QwQ-32B | 756.06 | 486.48 | 14.73 + 426.88 |
> | DeepSeek-R1-Distill-Llama-70B | 777.66 | 527.39 | 14.73 + 466.41 |
>
> Planning contributes only **3–6%** of end-to-end latency, while the adaptive budgeting reduces total latency by **34–42%** compared to vanilla decoding, yielding a net efficiency gain.

---

> ### Author Response · Authors · 2025-11-26
> **Rebuttal Part II**
>
> **3. Reliability of the Planner LLM's Complexity Estimation**
>
> This is an excellent question regarding the weak standalone performance of our lightweight planner (LLaMA-3.1-8B with $48.76\%$ MATH accuracy).
>
> - **Coarse Signal Suffices:** We find that the planner **does not need to solve the task well; it only needs to provide a coarse-grained, relative ranking of sub-question difficulty**.
> - **Stability Across Scales:** We conducted additional experiments using a much stronger planner (LLaMA-3.3-70B) (shown in **following tables**) and found that the overall $\mathcal{E}^3$ performance of Plan-And-Budget remained highly consistent. This confirms that even the smaller 8B model reliably produces the necessary complexity signal for effective budgeting.
> - **Decomposition is Key:** The planner's primary role is to provide a structured decomposition (the "scaffold") and a relative weighting, which focuses the main LLM's effort.
>
>
> Table 2: **MATH-500** Experiment results for **DeepSeek-R1-Distill-Qwen-32B**  (planning and budgeting by LLaMA 3.3-**70B**)
> |Methods | Acc(%) $\uparrow$ | Avg.Tok. $\downarrow$ | $\mathcal{E}^3$ $\uparrow$ | Percentage $\uparrow$|
> |-|-|-|-|-|
> | Vanilla|89.76 ± 0.26|2105.12 ± 31.94|3.83|100.00%|
> | Global Budget|89.60 ± 0.88|1526.15 ± 10.09|5.26|137.45%|
> | Planned Vanilla|90.12 ± 0.39|1633.00 ± 43.75|4.97|129.95%|
> | Planned Global Budget|89.64 ± 0.43|1377.95 ± 22.21|5.83|152.36%|
> | Plan-And-Budget **+Uniform**|89.44 ± 0.52|1319.23 ± 22.30|6.06|158.44%|
> | Plan-And-Budget **+Weighted**|89.40 ± 0.73|1320.31 ± 40.66|6.05|158.16%|
> | Plan-And-Budget **+Linear**|88.96 ± 0.59|1294.38 ± 48.75|6.11|159.75%|
> | Plan-And-Budget **+Exponential**|89.12 ± 0.58|1348.38 ± 37.14|5.89|153.90%|
> | Plan-And-Budget **+Polynomial**|89.76 ± 0.41|1263.16 ± 28.74|6.38|166.66%|
> | Plan-And-Budget **+Cosine**|89.16 ± 1.28|1304.74 ± 48.53|6.09|159.19%|
>
>
> Table 3: **MATH-500** Experiment results for **DeepSeek-R1-Distill-Llama-70B**  (planning and budgeting by LLaMA 3.3-**70B**)
> |Methods | Acc(%) $\uparrow$ | Avg.Tok. $\downarrow$ | $\mathcal{E}^3$ $\uparrow$ | Percentage $\uparrow$|
> |-|-|-|-|-|
> |Vanilla|90.44 ± 0.61|2286.63 ± 26.42|3.58|100.00%|
> |Global Budget|90.80 ± 0.62|1810.83 ± 51.64|4.55|127.28%|
> |Planned Vanilla|90.44 ± 0.71|1799.48 ± 36.42|4.55|127.07%|
> |Planned Global Budget|89.44 ± 0.67|1527.37 ± 18.04|5.24|146.42%|
> |Plan-And-Budget **+Uniform**|90.12 ± 0.63|1513.84 ± 43.93|5.36|149.98%|
> |Plan-And-Budget **+Weighted**|89.92 ± 0.30|1513.59 ± 60.18|5.34|149.34%|
> |Plan-And-Budget **+Linear**|89.76 ± 1.31|1480.78 ± 31.96|5.44|152.11%|
> |Plan-And-Budget **+Exponential**|90.04 ± 0.54|1472.01 ± 41.92|5.51|153.97%|
> |Plan-And-Budget **+Polynomial**|89.72 ± 0.58|1467.00 ± 35.92|5.49|153.40%|
> |Plan-And-Budget **+Cosine**|90.00 ± 0.62|1462.90 ± 43.88|5.54|154.79%|
>
>
> Table 4: **TravelPlanner** Experiment results for **DeepSeek-R1-Distill-Qwen-32B**  (planning and budgeting by LLaMA 3.3-**70B**)
> |Methods | Acc(%) $\uparrow$ | Avg.Tok. $\downarrow$ | $\mathcal{E}^3$ $\uparrow$ | Percentage $\uparrow$|
> |-|-|-|-|-|
> |Vanilla|14.33 ± 2.17|1430.14 ± 43.73|0.14|100.00%|
> |Global Budget|13.78 ± 1.20|1158.81 ± 20.23|0.16|114.12%|
> |Planned Vanilla|21.22 ± 2.56|1379.60 ± 58.31|0.33|227.31%|
> |Planned Global Budget|21.44 ± 1.65|1215.41 ± 36.19|0.38|263.40%|
> |Plan-And-Budget **+Uniform**|23.11 ± 2.47|1237.61 ± 47.96|0.43|300.54%|
> |Plan-And-Budget **+Weighted**|22.89 ± 3.48|1208.54 ± 68.78|0.43|301.94%|
> |Plan-And-Budget **+Linear**|21.89 ± 2.53|1241.55 ± 16.59|0.39|268.79%|
> |Plan-And-Budget **+Exponential**|22.22 ± 1.11|1182.30 ± 36.35|0.42|290.83%|
> |Plan-And-Budget **+Polynomial**|23.44 ± 2.82|1194.25 ± 56.38|0.46|320.41%|
> |Plan-And-Budget **+Cosine**|22.22 ± 2.00|1164.77 ± 39.57|0.42|295.21%|
>
>
> Table 5: **TravelPlanner** Experiment results for **DeepSeek-R1-Distill-Llama-70B**  (planning and budgeting by LLaMA 3.3-**70B**)
> |Methods | Acc(%) $\uparrow$ | Avg.Tok. $\downarrow$ | $\mathcal{E}^3$ $\uparrow$ | Percentage $\uparrow$|
> |-|-|-|-|-|
> |Vanilla|26.22 ± 1.82|1361.37 ± 47.93|0.50|100.00%|
> |Global Budget|24.33 ± 2.30|1215.29 ± 35.05|0.49|96.45%|
> |Planned Vanilla|33.11 ± 3.39|1392.14 ± 17.92|0.79|155.94%|
> |Planned Global Budget|34.22 ± 0.84|1248.56 ± 49.10|0.94|185.72%|
> |Plan-And-Budget **+Uniform**|30.44 ± 2.02|1149.63 ± 32.45|0.81|159.60%|
> |Plan-And-Budget **+Weighted**|30.22 ± 2.44|1202.07 ± 32.48|0.76|150.44%|
> |Plan-And-Budget **+Linear**|30.00 ± 1.11|1162.22 ± 45.53|0.77|153.34%|
> |Plan-And-Budget **+Exponential**|31.56 ± 2.13|1156.43 ± 30.26|0.86|170.56%|
> |Plan-And-Budget **+Polynomial**|33.00 ± 3.03|1142.54 ± 25.70|0.95|188.74%|
> |Plan-And-Budget **+Cosine**|31.00 ± 3.78|1115.86 ± 22.65|0.86|170.54%|
>
> We will report these results in the appendix.

---

> ### Author Response · Authors · 2025-11-26
> **Rebuttal Part III**
>
> **4. Contribution of Planning vs. Budgeting**
>
> We agree that structural decomposition (Planning) inherently provides benefits, which is evident in the gains from *Planned Vanilla* versus *Vanilla*. This validates our first principle: **reasoning should be structured.**
>
> However, the addition of *Local Budgeting* yields substantial and non-trivial additional gains, validating our second principle: **computation should be adaptive.**
> - **Example (TravelPlanner, DS-LLaMA-70B):**
>     - Planned Vanilla $\rightarrow$ $\mathcal{E}^3$ of $0.64$ (Planning-only gain)
>     - Planned Global Budget $\rightarrow$ $\mathcal{E}^3$ of $0.77$
>     - Plan-And-Budget (+Polynomial) $\rightarrow$ $\mathcal{E}^3$ of $0.93$ (Additional $45\%$ gain over Planned Vanilla)
> - **Conclusion:** Planning provides the scaffold; **budgeting provides the final, adaptive mechanism** that aligns inference effort with evolving needs, yielding the highest efficiency gains overall.
>
> **5. On the $\mathcal{E}^3 = A^2/T$ Metric**
>
> The choice of $A^2/T$ is deliberate and necessary to mitigate **reward hacking**, where a conventional metric like $A/T$ is maximized by degenerate strategies (e.g., answering only a handful of very simple questions correctly for minimal $T$, yielding a high $A/T$ but low overall $A$).
>
> The $\mathcal{E}^3$ metric is designed to:
> - **Prioritize Correctness:** The quadratic factor heavily penalizes accuracy drops, ensuring the method aims for **efficient correct reasoning**, not simply **fast token generation**.
> - **Differentiate Top Performers:** On the **NaturalInstructions** task, where multiple methods show similar $A/T$, $\mathcal{E}^3$ effectively differentiates them based on their quality, correctly identifying the best-performing approach (which typically has the highest $\mathcal{E}^3$).
>
> Crucially, we confirm that our conclusions remain robust when using $A/T$; the relative ranking of Plan-and-Budget over baselines is stable across both metrics. We will include the following tables comparing $A/T$ and $\mathcal{E}^3$ in the appendix.
>
> Table: Experiment results comparing $\mathcal{E}^3$ and $A/T$ on **MATH-500**.
> |Models→ | DeepSeek-R1-Distill-Qwen-32B ($\mathcal{E}^3$/$A/T$) | QwQ-32B ($\mathcal{E}^3$/$A/T$) | DeepSeek-R1-Distill-Llama-70B ($\mathcal{E}^3$/$A/T$) | o4-mini ($\mathcal{E}^3$/$A/T$) |
> |-|-|-|-|-|
> | Vanilla|3.83/4.26|2.04/2.41|3.58/3.96|12.20/13.10|
> | Global Budget|5.26/5.87|3.20/3.53|4.55/5.01|13.25/14.43|
> | Planned Vanilla|4.40/4.83|2.23/2.60|4.20/4.56|15.65/17.04|
> | Planned Global Budget|5.36/5.88|2.91/3.30|5.16/5.57|14.39/15.67|
> | Plan-And-Budget **+Uniform**|5.64/6.26|3.28/3.70|5.41/5.86|15.88/17.38|
> | Plan-And-Budget **+Weighted**|5.51/6.09|3.08/3.53|5.51/5.95|15.60/17.03|
> | Plan-And-Budget **+Linear**|6.07/6.74|3.31/3.76|5.57/6.03|15.34/16.94|
> | Plan-And-Budget **+Exponential**|5.93/6.53|3.33/3.79|5.89/6.33|15.72/17.29|
> | Plan-And-Budget **+Polynomial**|5.91/6.56|3.32/3.76|5.58/6.07|15.55/17.21|
> | Plan-And-Budget **+Cosine**|5.92/6.58|3.40/3.84|5.80/6.24|15.95/17.46|
>
> Table: Experiment results comparing $\mathcal{E}^3$ and $A/T$ on **NaturalInstructions**.
> |Models→ | DeepSeek-R1-Distill-Qwen-32B ($\mathcal{E}^3$/$A/T$) | QwQ-32B ($\mathcal{E}^3$/$A/T$) | DeepSeek-R1-Distill-Llama-70B ($\mathcal{E}^3$/$A/T$) | o4-mini ($\mathcal{E}^3$/$A/T$) |
> |-|-|-|-|-|
> | Vanilla|1.95/4.49|1.02/2.37|2.08/4.82|4.84/10.25|
> | Global Budget|2.33/5.44|1.47/3.29|2.48/5.67|4.88/10.75|
> | Planned Vanilla|2.10/4.93|1.13/2.61|2.29/5.28|5.56/12.69|
> | Planned Global Budget|2.52/5.92|1.55/3.50|2.61/6.14|5.34/12.20|
> | Plan-And-Budget **+Uniform**|2.61/6.36|1.75/3.99|2.78/6.47|5.57/12.64|
> | Plan-And-Budget **+Weighted**|2.57/6.22|1.65/3.80|2.96/6.87|5.14/11.76|
> | Plan-And-Budget **+Linear**|2.73/6.56|1.72/3.97|2.88/6.86|5.37/12.16|
> | Plan-And-Budget **+Exponential**|2.64/6.37|1.70/3.92|2.93/6.86|5.23/11.97|
> | Plan-And-Budget **+Polynomial**|2.86/6.91|1.70/3.87|2.91/6.73|5.48/12.29|
> | Plan-And-Budget **+Cosine**|2.73/6.60|1.69/3.86|2.79/6.51|5.42/12.22|
>
> Table: Experiment results comparing $\mathcal{E}^3$ and $A/T$ on **TravelPlanner**.
> |Models→ | DeepSeek-R1-Distill-Qwen-32B ($\mathcal{E}^3$/$A/T$) | QwQ-32B ($\mathcal{E}^3$/$A/T$) | DeepSeek-R1-Distill-Llama-70B ($\mathcal{E}^3$/$A/T$) | o4-mini ($\mathcal{E}^3$/$A/T$) |
> |-|-|-|-|-|
> | Vanilla|0.14/1.00|0.35/1.02|0.50/1.93|0.086/0.74|
> | Global Budget|0.16/1.19|0.37/1.22|0.49/2.00|0.056/0.67|
> | Planned Vanilla|0.30/1.50|0.38/1.01|0.64/2.09|0.091/0.74|
> | Planned Global Budget|0.41/1.82|0.39/1.10|0.77/2.51|0.037/0.52|
> | Plan-And-Budget **+Uniform**|0.35/1.68|0.45/1.26|0.81/2.56|0.090/0.82|
> | Plan-And-Budget **+Weighted**|0.45/1.91|0.40/1.19|0.74/2.48|0.088/0.81|
> | Plan-And-Budget **+Linear**|0.34/1.72|0.45/1.29|0.86/2.72|0.104/0.89|
> | Plan-And-Budget **+Exponential**|0.40/1.85|0.46/1.30|0.86/2.69|0.075/0.76|
> | Plan-And-Budget **+Polynomial**|0.47/2.01|0.49/1.39|0.93/2.85|0.104/0.91|
> | Plan-And-Budget **+Cosine**|0.36/1.77|0.52/1.43|0.85/2.70|0.077/0.78|

---

> ### Author Response · Authors · 2025-11-26
> **Rebuttal Part IV**
>
> **6. Comparison to Confidence-Based Early Stopping**
>
> This is an excellent baseline suggestion. We implemented a confidence-based early termination baseline following recent work in the literature, Certaindex [2]. The results confirm the necessity of explicit planning and local allocation:
>
> [2] Fu, Yichao, et al. "Efficiently serving llm reasoning programs with certaindex." arXiv e-prints (2024): arXiv-2412.
>
> Table: The average **accuracy(%) / token / $\mathcal{E}^3$** comparison.
> | | Vanilla | Global Budget | Plan-And-Budget | Certaindex [2] |
> | - | - | - | - | - |
> | DeepSeek-R1-Distill-Qwen-32B | 89.76±0.26 / 2105.12±31.94 / 3.83 | 89.60±0.88 / 1526.15±10.09 / 5.26 | **90.04±0.46 / 1336.27±31.18 / 6.07** | 81.44±1.38 / 1985.57±46.12 / 3.34 |
> | QwQ-32B | 84.88±1.18 / 3523.72±97.42 / 2.04 | **90.56±0.33** / 2565.18±37.10 / 3.20 | 88.60±0.28 / **2306.83±24.11 / 3.40** | 86.12±0.73 / 5141.72±82.95 / 1.44 |
> | DeepSeek-R1-Distill-Llama-70B | 90.44±0.61 / 2286.63±26.42 / 3.58 | 90.80±0.62 / 1810.83±51.64 / 4.55 | **93.04±0.22 / 1469.29±73.77 / 5.89** | 81.96±1.24 / 2200.68±79.88 / 3.05 |
>
> The confidence-based approach prematurely terminates reasoning steps, sacrificing accuracy (down to $\approx 81\%$), and often fails to significantly reduce tokens compared to our method, leading to a much lower $\mathcal{E}^3$. **Our approach, which adapts before generation via planning and budget, is significantly more effective** than reactively stopping generation based on confidence.
>
> **7. Other Questions (Modest Gains, Sensitivity, BAM vs. Heuristics)**
>
> - **Modest Gains (NaturalInstructions):** We agree the gains are smaller on NaturalInstructions. This is because it is less prone to overthinking/underthinking, and the tasks are often shorter and more structured. The potential for token savings is inherently limited in this domain.
> - **Sensitivity to Number of Sub-Questions (2-5):** The performance is not overly sensitive within the 2–5 range (which is controlled by the planner prompt). The main LLM's adaptive capabilities handle minor variations. However, forcing the planner to create too many sub-questions moves the weak planner (8B) from simple decomposition to complex, detailed problem-solving, which harms accuracy. This further validates our design of using short, high-level decompositions. We will include a small sensitivity table.
> - **BAM vs. Heuristic Decay Schedules:** Polynomial decay (and other decay families) are heuristic implementations inspired by BAM, not direct optimizers. Because $c_{ij}$ and $\beta_{ij}$ are not observable in black-box LLMs, we implement practical approximations that follow BAM’s structural insights (e.g., diminishing returns, monotonicity).

---

### Author Response · Authors · 2025-12-02
**General Rebuttal Summary**

Dear Reviewers and Area Chairs,

We sincerely thank all reviewers for their valuable time and insightful comments on our work. We have responded and addressed their concerns individually. We also include additional experimental results to strengthen our claims.

**Recognized Strengths:**

We are grateful that reviewers acknowledged both **novelty and thoroughness**:
- **R3 (fET4), R4 (Mzjt), R2 (P17L), R5 (f6TR)** praised the method design as **rigorous** and **elegant** and appreciated the strong **generalizability** demonstrated by comprehensive experiments across four models (DS-Qwen-32B, QwQ-32B, DS-LLaMA-70B, o4-mini) and three diverse reasoning domains.
- The core idea of dynamically allocating token budgets to sub-questions based on **uncertainty** is recognized as **novel and principled**.
- **R1 (V5hD) and R2 (P17L)** appreciated the **theoretical framework (BAM)** and the $\mathcal{E}^3$ metric design as appropriate tools for addressing the timely and important challenge of balancing accuracy and efficiency in LLM inference.
- **R4 (Mzjt)** highlighted the significant result that Plan-And-Budget effectively serves as an **inference-time equalizer**, helping smaller models (DS-Qwen-32B, $\mathcal{E}^3=0.47$) match the efficiency of larger models (DS-LLaMA-70B, $\mathcal{E}^3=0.50$).

**Addressing Reviewer Concerns:**

Q1: Is it limited to decay-based scheduling? Why is budgeting necessary over planning alone? [R1 (V5hD), R3 (fET4), R2 (P17L), R5 (f6TR)]

**A1: Our method is flexible, and adaptive budgeting provides essential, non-trivial gains over planning alone.**
- Our method supports a broad family of allocation strategies: **uniform, weighted** (non-decay, proportional to difficulty), and decay-based (linear, polynomial, exponential, cosine).
- For tasks requiring **rich generative output** (e.g., NaturalInstructions), the weighted or uniform strategies perform better, aligning with the observed need for balanced reasoning effort.
-  Planning provides the structural scaffold, but adaptive budgeting yields significant additional efficiency. On TravelPlanner (DS-LLaMA-70B), simply adding local budgeting boosts $\mathcal{E}^3$ from 0.79 (Planned Vanilla) to **0.95** (Plan-And-Budget), achieving an 18% token reduction and demonstrating that the adaptive allocation is critical for maximizing efficiency.

Q2: Computational cost of planning, and end-to-end latency? [R1 (V5hD), R3 (fET4), R2 (P17L)]

**A2: The planning overhead is negligible and fully amortized by the token savings.**

- The planner runs once per query, producing short sub-questions in a single LLM call. Planning + budget allocation step constitutes only **3–6%** of the total end-to-end inference latency.
- Due to the substantial token savings in the reasoning phase, the total latency for Plan-And-Budget is **34–42%** lower than vanilla and **7–10%** lower than global budget baselines across all models, confirming the framework reduces total inference cost.

Q3: Why is explicit budgeting necessary instead of reactive methods like Early Termination? [R5 (f6TR)]

**A3: Reactive early termination fails to solve the root problem of miscalibration.**

- **Proactive vs. Reactive:** Early termination is a reactive method that only prunes an overlong chain after tokens have been wasted. It cannot redistribute computation.
- **Superior Performance:** We conducted a direct comparison to an early termination baseline [1] and found that it significantly sacrificed accuracy (down to 81% on MATH-500) while failing to achieve the token savings of Plan-And-Budget (2000 tokens vs. **1336** tokens).

Q4: The "Theory–Practice Gap" - How can we justify BAM if $c_{ij}$ and $\beta_{ij}$ aren't used? [R1 (V5hD), R5 (f6TR)]
**A4: BAM provides a normative foundation, and heuristics are empirically validated proxies for the optimal shape.**

- **BAM's Role:** The **Budget Allocation Model (BAM)** is not a literal runtime algorithm, but a theoretical framework used to derive the structural principles of optimal resource allocation (e.g., diminishing returns, unimodality of effort).
- **Practical Bridge:** The decay schedules are **principled, low-cost heuristics** that mimic the predicted optimal shape (front-loaded emphasis where epistemic uncertainty is highest), without the computational overhead of parameter estimation.
- **Validation:** Our **Budget Reallocation** experiment, which uses dynamic signals to reallocate budget on-the-fly, shows further performance gains (up to **+5.4%** $\mathcal{E}^3$ over fixed decay), confirming that the underlying BAM intuition about adaptive computation is correct and highly valuable.

We sincerely appreciate the constructive comments and hope these additional results and clarifications address any remaining concerns and fully demonstrate the practical value and theoretical grounding of our work.

[1] Fu, Yichao, et al. "Efficiently serving llm reasoning programs with certaindex." arXiv e-prints (2024): arXiv-2412.

---

### Meta-Review · Area_Chair_BBcS · 2026-01-09

**Summary:**

The paper presents a novel conceptual framework and an associated approximated implementation for deciding when to stop thinking and produce an answer in thinking LLMs.

The paper received a 2, a 4, and three 8s. Most reviewers praise the paper for addressing a real problem, providing some theoretical grounding, and providing a very strong methodology with rigorous experimental results.

I carefully looked at the critiques. The reviewer who gave a 4 made a lot of critiques. The high level critiques are (and this point is raised by many others) that conceptual framework and actual implementation are different. Authors argue that conceptual framework only guides... and the algorithm was designed based on practical considerations. I understand the argument, though am not satisfied by it. Eventually, a framework must ~reflect the algorithm closely or it feels like an afterthought. Still, given the positives in everything else, I am OK to not make it a strong deterrant in accepting the paper. Other issues were low level: whether ROUGE was normalized or not, ablations on planner, requesting for cost-comparisons since #tokens differ across models, etc. The authors gave very strong responses, with detailed additional experiments to satisfy the points.

The reviewer who gave a 2 primarily asked for comparison with another paper (and questioned novelty). The authors distinguished themselves conceptually as well as experimentally in response, and showed strong results against this related work. The authors _must_ add this comparison in the revised version, but the question gets answered well. The other questions asks specific questions about the design of the algorithm (such as why adaptive budgeting is necessary, or whether the task can be reliably decomposed)... for which I felt that the author responses were adequate.

Overall, I feel that the paper is strong and can be accepted.

**Reviewer Concerns:**

see above. main issue is gap between theory and algorithm. I hope that the authors underplay the theory and describe where they get guided by it in the algorithm

**Reviewer Scores:**

4 would likely have become 6. 2 should have improved also... maybe to 4 or 5. 8s would likely not have gone down

---

### Decision · Program_Chairs · 2026-01-26

Accept (Poster)